# Gjd2b-mediated gap junctions promote glutamatergic synapse formation and dendritic elaboration in Purkinje neurons

Sahana Sitaraman[†], Gnaneshwar Yadav[†], Vandana Agarwal, Shaista Jabeen, Shivangi Verma, Meha Jadhav, Vatsala Thirumalai*

National Centre for Biological Sciences, Tata Institute of Fundamental Research, Bangalore, India

**Abstract** Gap junctions between neurons serve as electrical synapses, in addition to conducting metabolites and signaling molecules. During development, early-appearing gap junctions are thought to prefigure chemical synapses, which appear much later. We present evidence for this idea at a central, glutamatergic synapse and provide some mechanistic insights. Loss or reduction in the levels of the gap junction protein Gjd2b decreased the frequency of glutamatergic miniature excitatory postsynaptic currents (mEPSCs) in cerebellar Purkinje neurons (PNs) in larval zebrafish. Ultrastructural analysis in the molecular layer showed decreased synapse density. Further, mEPSCs had faster kinetics and larger amplitudes in mutant PNs, consistent with their stunted dendritic arbors. Time-lapse microscopy in wild-type and mutant PNs reveals that Gjd2b puncta promote the elongation of branches and that CaMKII may be a critical mediator of this process. These results demonstrate that Gjd2b-mediated gap junctions regulate glutamatergic synapse formation and dendritic elaboration in PNs.

**\*For correspondence:**
vatsala@ncbs.res.in

[†]These authors contributed equally to this work

## Introduction

The formation of synapses is an elaborate multi-step process involving several classes of signaling molecules and electrical activity (*Waites et al., 2005*; *McAllister, 2007*). Several studies support the view that connections that favor correlated activity between presynaptic and postsynaptic neurons are strengthened while those that are poorly correlated are weakened and eliminated (*Katz and Shatz, 1996*; *Kirkby et al., 2013*). Correlated activity promotes dendritic elaboration (*Wong and Ghosh, 2002*; *Parrish et al., 2007*) and the subsequent formation of synaptic sites on those newly formed arbors (*Niell et al., 2004*). However, we understand little about the molecular mechanisms linking correlated activity to dendritic arbor elaboration and synaptogenesis (*Sin et al., 2002*; *Redmond and Ghosh, 2005*; *Schwartz et al., 2009*; *Chen et al., 2012*).

Electrical synapses, formed via gap junctions between connected neuronal pairs, allow the passage of ions, metabolites, and second messengers, and are ideally suited for enhancing correlations in activity between connected neurons (*Pereda, 2014*; *Connors, 2017*). Indeed, several lines of evidence suggest that gap junctions play critical roles in circuit assembly. First, neurons show increased gap junctional connectivity early on in development at stages that precede chemical synapse formation (*Montoro and Yuste, 2004*; *Marin-Burgin et al., 2008*; *Jabeen and Thirumalai, 2013*). Second, knocking out or knocking down gap junction proteins at these stages results in decreased chemical synapse connectivity at later stages (*Maher et al., 2009*; *Todd et al., 2010*). Gap junctions could mediate chemical synaptogenesis by increasing correlations in activity, transmitting synaptogenic signaling molecules, or providing enhanced mechanical stability at junctional sites between connected pairs. It is not clear which of these functions of gap junctions are critical for synaptogenesis. In addition, while the role of excitatory chemical synapses in sculpting neuronal arbors has been

investigated in several circuits (*Inglis et al., 2002*; *Haas et al., 2006*; *Cline and Haas, 2008*), little is known regarding such a role for electrical synapses. We set out to investigate whether gap junctions regulate structural and functional synaptic development of cerebellar Purkinje neurons (PNs), and if yes, what mechanisms may be involved using larval zebrafish as our model system.

The cerebellum is critical for maintaining balance and for coordination of movements. It is one of the most primitive organs of the vertebrate central nervous system, has a layered structure, and its circuitry is conserved from fish to mammals (*Nieuwenhuys, 1967*). PNs are principal output neurons of the cerebellar cortex and receive excitatory and inhibitory synaptic inputs on their elaborate dendritic arbors. PNs receive thousands of glutamatergic inputs from parallel fiber axons of granule cells and relatively fewer inhibitory inputs from molecular layer interneurons. They also receive strong excitatory inputs from inferior olivary climbing fiber (CFs) axons on their proximal dendrites. In zebrafish, PNs are specified by 2.5 days post fertilization (dpf), begin elaborating dendritic arbors soon after, and a distinct molecular layer consisting of PN dendritic arbors becomes visible by 5 dpf (*Bae et al., 2009*; *Hamling et al., 2015*). In addition, excitatory and inhibitory synaptic currents can be recorded in PNs of 4 dpf zebrafish larvae, evidencing a nascent functional circuit at this stage (*Sengupta and Thirumalai, 2015*).

The gap junction delta 2b protein (Gjd2b), also referred to as Connexin 35b or 35.1 (Cx35/Cx35.1), is the teleostean homolog of the mammalian Cx36, which is the predominant neural gap junction protein. PNs begin to express Gjd2b by about 4 dpf, and the level of expression increases steadily at least until 15 dpf (*Jabeen and Thirumalai, 2013*). We asked whether Gjd2b regulates structural and functional synaptic development of PNs. Using knockdown and knockout approaches, we show here that Gjd2b is indeed required for the formation of glutamatergic synapses and for normal dendritic arbor growth of PNs.

## Results

### Expression and knockdown of Gjd2b in PNs using morpholinos

Gjd2b is expressed at high levels in the cerebellum of larval zebrafish beginning at 4 dpf, a stage at which cerebellar neurons have been specified but chemical synaptic connections are still forming (*Bae et al., 2009*; *Jabeen and Thirumalai, 2013*; *Figure 1—figure supplement 1A*). Using an antibody that recognizes both Gjd2a and Gjd2b, we confirmed that Gjd2a/b puncta localized to PN cell membrane in their cell bodies and dendrites (*Figure 1—figure supplement 1B*). To test whether Gjd2b mediates chemical synaptogenesis in PNs, we knocked it down with a splice-blocking morpholino antisense oligonucleotide (referred to as Gjd2b-MO) targeted to the splice junction between exon 1 and intron 1 of gjd2b (*Figure 1—figure supplement 2A*, top). Injection of this splice blocking morpholino (Gjd2b-MO) into 1–4 cell stage embryos interferes with normal splicing of the *gjd2b* gene product, resulting in the inclusion of a 40 base intronic segment in the mature mRNA (*Figure 1—figure supplement 2C*, left). The mis-spliced gene product was detected in 2 dpf and 5 dpf larvae (*Figure 1—figure supplement 2B*) and the sequence reveals a premature stop codon resulting in a putative truncated protein of 24 amino acid residues in the N-terminus (*Figure 1—figure supplement 2C*, right). Gjd2a/b immunoreactivity in the PN and molecular layers of the cerebellum is reduced in Gjd2b-MO larvae compared to uninjected larvae (*Figure 1—figure supplement 2D*, Mann–Whitney test, p<0.001) confirming effective knockdown. Injection of a morpholino with a 5-base mismatch (CTRL) did not alter Gjd2a/b protein levels (*Figure 1—figure supplement 2E*, Mann–Whitney test, p=0.14).

### Morphant PNs exhibit deficits in AMPAR-mediated synaptic transmission

We followed physiological changes induced by Gjd2b loss by recording AMPAR-mediated miniature excitatory postsynaptic currents (mEPSCs) in PNs of uninjected and Gjd2b-MO larvae using previously established methods (*Sengupta and Thirumalai, 2015*). mEPSCs in morphants occurred less frequently than in uninjected larvae (*Figure 1A*), reflected as increased inter-event intervals in the morphants (*Figure 1C*, Mann–Whitney test, U = 37,285, p<0.0001). mEPSCs in morphants also showed a small increase in the peak amplitude (*Figure 1D*, Mann–Whitney test, U = 25,537, p=0.002) and faster decay time constants (*Figure 1B, F*, Mann–Whitney test, U = 25,018, p=0.014).

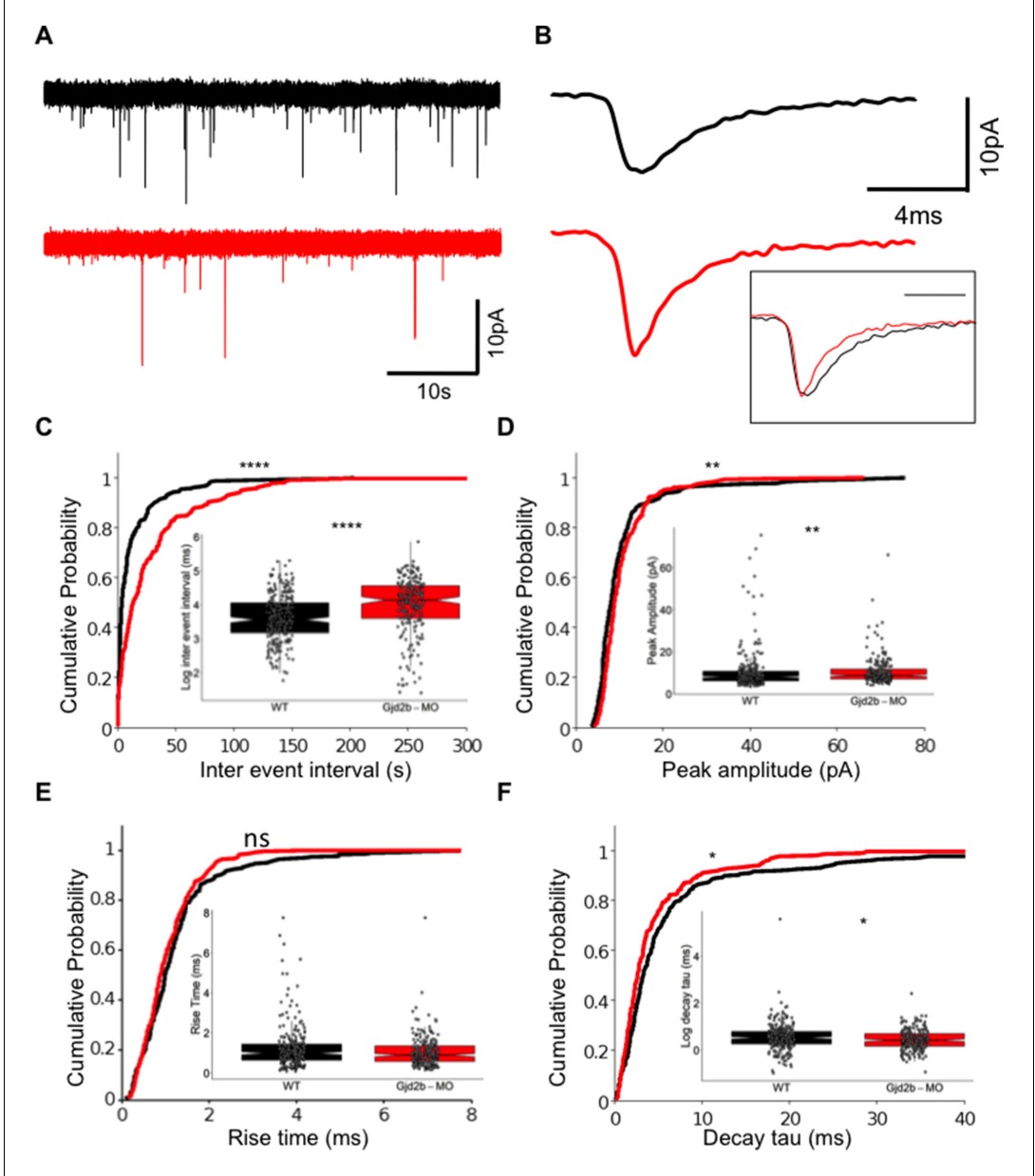

**Figure 1.** Knocking down Gjd2b reduces glutamatergic miniature excitatory postsynaptic current (mEPSC) frequency in Purkinje neurons (PNs) by potentially decreasing synaptic number. (A) Raw traces of mEPSC recordings from PNs in control (black) and Gjd2b-MO-injected (red) larvae. (B) Average mEPSC waveforms from control and Gjd2b-MO-injected larvae. Inset: scaled mEPSC to show faster decay time of mEPSCs in morphants. Neurons were held at −65 mV. (C–F) Cumulative probability distributions and boxplots of mEPSC inter-event intervals (C), peak amplitudes (D), 10–90% rise times (E), and decay tau (F) in control (black) and Gjd2b-MO-injected (red) larvae. N = 7 cells from 7 larvae from 5 clutches for control and 12 cells from 12 larvae from 8 clutches for the Gjd2b-MO group. **p<0.01; ***p<0.001; ****p<0.0001; Mann–Whitney U test. See also *Figure 1—figure supplements 1* and *2*. Data used for quantitative analyses are available in *Figure 1—source data 1*.

The online version of this article includes the following source data and figure supplement(s) for figure 1:

**Source data 1.** mEPSC data.

**Figure supplement 1.** Connexin 35 (Cx35) is expressed in the cerebellum in Purkinje neurons (PNs).

**Figure supplement 2.** Effective knockdown of Gjd2b after injection of splice-blocking morpholino.

**Figure supplement 2—source data 1.** Fluorescence intensity data.

The increase in inter-event intervals of mEPSCs suggests that knocking down Gjd2b leads to a decrease in the density of synapses impinging on PNs.

## Generation of *gjd2b*<sup>-/-</sup> zebrafish using TALENs

As an independent but stable approach (*Kok et al., 2015*; *Stainier et al., 2017*) to verify effects of a complete loss of Gjd2b on glutamatergic synapses, we generated *gjd2b*<sup>-/-</sup> fish using TALENS (*Christian et al., 2010*) targeted against exon 1 of *gjd2b* (*Figure 2A*). We isolated several alleles with indels in the target region and chose one allele, *gjd2b*<sup>ncb215</sup> (hereinafter referred to as *gjd2b*<sup>-/-</sup>), for further analysis. In this allele, insertion of a single G after the translation start site abolished an *Xho*I restriction enzyme recognition site within the target region (*Figure 2B*) and resulted in failure of *Xho*I restriction digestion in homozygotes and partial digestion in heterozygotes (*Figure 2C*). This single-nucleotide insertion caused a frame shift and insertion of a premature stop codon, with a predicted truncated protein of 55 amino acid residues (*Figure 2D*). Homozygotes with this mutation showed reduced Gjd2b-like immunoreactivity in their cerebellum (*Figure 2E, F*).

## Impaired AMPAR-mediated synaptic transmission in PNs of gjd2b<sup>-/-</sup> larvae

We recorded mEPSCs from PNs in gjd2b<sup>-/-</sup> larvae to determine if loss of Gjd2b leads to reduced glutamatergic synaptic contacts, as observed in the morphants (*Figure 1*). These recordings were performed with cesium gluconate internal solution, and under these conditions, the input resistance of mutant neurons was significantly higher compared to wild type (WT; *Figure 3—figure supplement 1B*). *gjd2b*<sup>-/-</sup> larvae showed a significant increase in mEPSC inter-event intervals (*Figure 3A, C*, Mann–Whitney test, U = 4906, p=0.02), recapitulating the results observed after knock down of Gjd2b with Gjd2b-MO (*Figure 1C*). Further, homozygous mutants also showed an increase in peak amplitudes (*Figure 3A, B, D*, Mann–Whitney test, U = 3385, p<0.0001), and faster kinetics as revealed by decreases in rise times (*Figure 3B, E*, Mann–Whitney test, U = 2322, p<0.0001) and decay time constants (*Figure 3B, F*, Mann–Whitney test, U = 2791, p<0.0001). These results are consistent with the results obtained after knocking down Gjd2b with Gjd2b-MO and point to a decrease in the number of glutamatergic synaptic contacts impinging on PNs.

However, this change may also be due to a decrease in presynaptic probability of transmitter release or an increase in the number of N-methyl-D-aspartate receptor (NMDAR)-only 'silent' synapses (*Liao et al., 1995*; *Wu et al., 1996*). To test if these possibilities are likely, we recorded synaptic currents evoked in WT PNs after stimulation of CFs (*Figure 4A*). These recordings were performed with potassium gluconate internal solution, and under these conditions, the input resistance of mutant neurons was comparable to WT (*Figure 3—figure supplement 1A*). As is the case in developing mammalian PNs (*Piochon et al., 2007*), we recorded no NMDAR-mediated component of the evoked EPSC in PNs (*Figure 4B*), suggesting that the decrease in mEPSC frequency after Gjd2b knockdown and knockout is likely not due to silent synapses. To further strengthen our results, we performed voltage clamp recordings from WT and *gjd2b*<sup>-/-</sup> PNs after administering tetrodotoxin (TTX) in the bath and brief puffs of 10 mM NMDA close to the soma. The cells were held at a depolarized potential (−20 mV) to remove the Mg<sup>2+</sup> block from NMDARs. No current response was observed after application of NMDA (*Figure 4C*), suggesting the absence of NMDARs on PNs in 7 dpf zebrafish larvae.

Secondly, we measured paired pulse ratios (PPRs) as an indicator of changes in presynaptic vesicle release probability. The CF-PN synapse shows paired pulse depression when pulses are placed 35 ms apart (*Figure 4D, F*). We tested a range of inter-stimulus intervals (ISIs) from 30 ms till 550 ms (in morphants) and till 1000 ms (in mutants) and found that the PPR varied as a function of the ISI but did not vary significantly between the uninjected and Gjd2b-MO groups (*Figure 4E*, two-way repeated-measures ANOVA, F = 3.979, df = 1, p=0.081) or WT and *gjd2b*<sup>-/-</sup> groups (*Figures 4G*, two-way repeated-measures ANOVA, F = 2.039, df = 1, p=0.17).

Taken together, these results suggest that knocking down or knocking out Gjd2b leads to a decrease in the number of glutamatergic synapses impinging on PNs. In subsequent experiments, we investigate this phenomenon in greater detail using the *gjd2b*<sup>-/-</sup> mutants.

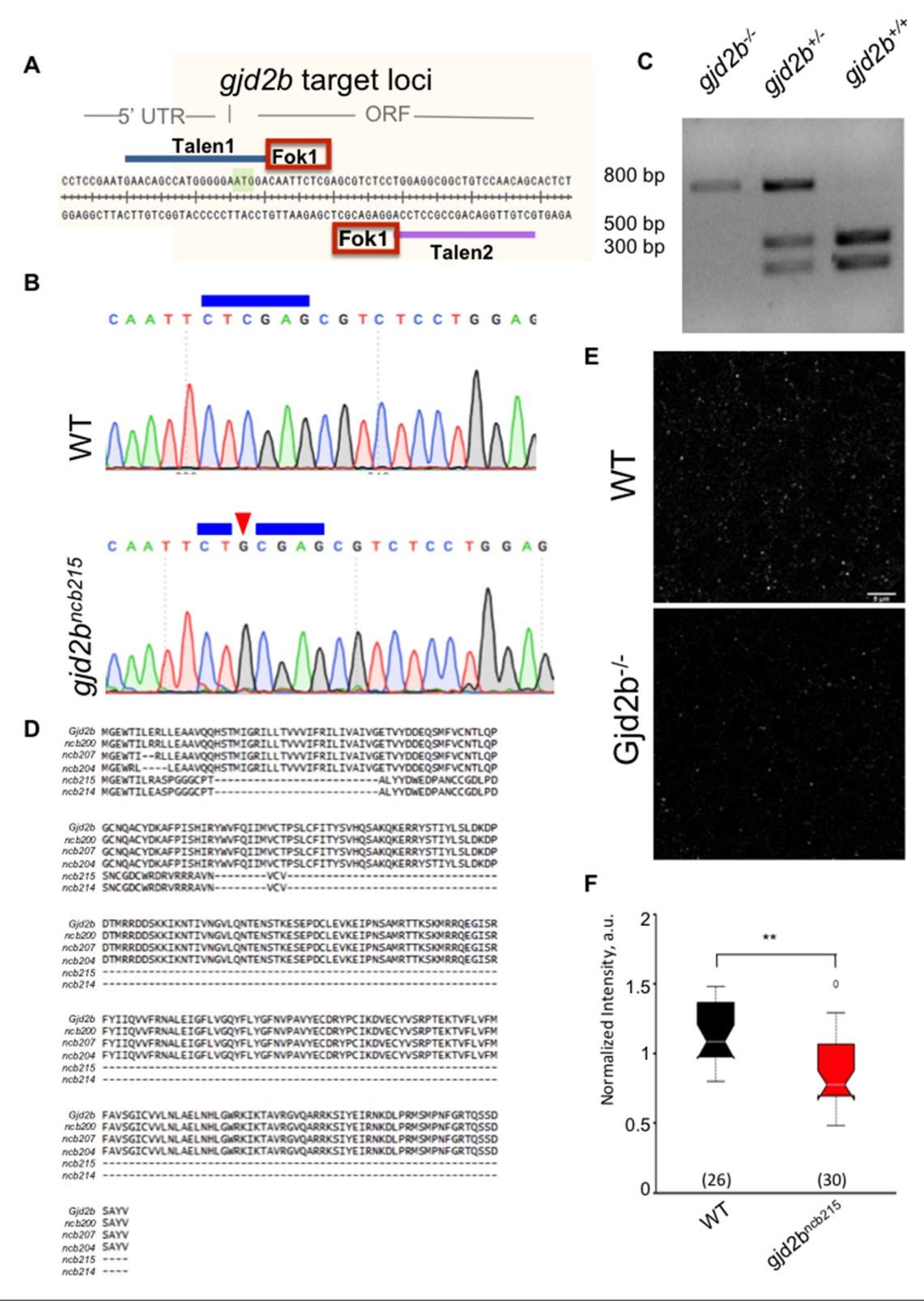

**Figure 2.** Generation of gjd2b mutant zebrafish. (**A**) Genomic region around the start codon (green box) of gjd2b gene was selected for TALEN design, where TALEN-1 recognition sequence (blue line) spans the 5'UTR, the start codon sequence, and a few nucleotides after the start codon. The TALEN-2 recognition sequence (purple line) begins after the 18 nucleotide spacer region. (**B**) Chromatograms of sequence read from wild-type (WT) and homozygous gjd2b[ncb215] fish generated in this study indicate an insertion of nucleotide G (red arrowhead) within the *Xho1* restriction site (blue line). (**C**)
*Figure 2 continued on next page*

*Figure 2 continued*

Representative gel image after *Xho1* restriction digestion analysis of an 800 bp amplicon (includes ~300 bp upstream and 500 bp downstream sequences from the point of insertion) shows undigested and partially digested bands in the homozygous and heterozygous mutants, respectively, compared to complete digestion in WT siblings. (D) Predicted amino acid sequences of various gjd2b mutant alleles generated in this study aligned with the WT Gjd2b sequence. gjd2b^ncb215 is predicted to code for the first six amino acid residues of Gjd2b followed by a nonsense sequence up to the 54th amino acid position. Presence of a premature stop codon terminates translation at this position. (E) Representative images of Gj2b-like immunoreactivity from WT and mutant fish. Staining was not completely abolished as the antibody also recognizes Gjd2a. (F) Gjd2b-like immunoreactivity is reduced in PNs of gjd2b^ncb215 compared to WT larvae (Mann–Whitney U test; ***p<0.001). Number of images analyzed is indicated in parentheses. Data used for quantitative analyses are available in *Figure 2—source data 1*.

The online version of this article includes the following source data for figure 2:

**Source data 1.** Fluorescence intensity data.

## Loss of Gjd2b reduces synapse density in the molecular layer

To finally confirm if the loss of Gjd2b indeed results in a decrease in synapse density, we quantified synapse density at the ultrastructural level. PNs send elaborate dendritic arbors into the molecular layer of the cerebellum where they make excitatory synapses with parallel fibers and CFs and inhibitory synapses with axons of molecular layer interneurons. Transmission electron micrographs (TEMs) were obtained from the molecular layer of the corpus cerebelli (CCe) of 7 dpf WT and mutant larvae from 60-nm-thick sections of the brain. Sections were taken at an interval of 1.2 μm to avoid oversampling the same synapses. Synapses were counted as membrane appositions with presynaptic vesicles on one side and electron-dense postsynaptic density on the other (*Figure 5A*). We found that the density of synapses per cubic micrometer was significantly lower in *gjd2b^-/-* larvae compared to WT larvae (*Figure 5B, E*, Mann–Whitney test, U = 86,268, p<0.0001). To understand if Gjd2b also regulates the maturation of synapses, we calculated the synapse maturation index (*Blue and Parnavelas, 1983*; *Haas et al., 2006*) for individual synaptic profiles, measured as the ratio of the area occupied by vesicles to the entire area of the presynaptic terminal (*Figure 5C*). The distribution of synapse maturation indices was not different between WT and mutant larvae (*Figure 5D, F*, Mann–Whitney test, U = 4922, p=0.12). In sum, these results indicate that Gjd2b is involved in the formation of synapses in the molecular layer of the cerebellum, to which PNs significantly contribute, but once formed, the maturation of synapses is independent of Gjd2b.

## Stunted dendritic arbors in gjd2b^-/- PNs

An increase in peak amplitude combined with faster kinetics of mEPSCs in *gjd2b^-/-* mutants (*Figure 3*) suggested that synapses are placed electrotonically closer to the soma in mutant PNs than in WT larvae. In addition, input resistance measured with potassium gluconate internal solution did not reveal any difference between WT and mutant PNs, while recordings with cesium gluconate showed that the input resistance was higher for mutant PNs (*Figure 3—figure supplement 1*), suggesting a dendritic leak conductance that was larger for WT neurons compared to mutant neurons. These data imply that loss of Gjd2b could result in stunted dendritic arbors and loss of distally located synapses. To understand if this is indeed the case, we labeled PNs of WT and mutant larvae in a mosaic fashion and imaged them daily from 5 dpf till 8 dpf (*Figure 6A, B*), a period when PNs in larval zebrafish are growing and making synaptic connections (*Bae et al., 2009*; *Hamling et al., 2015*).

We measured the total dendritic branch length (TDBL) of PNs in WT and mutant larvae at 5, 6, 7, and 8 dpf. WT PNs showed a significant increase in their TDBL from 5 dpf to 6 dpf, but not at later stages (*Figure 6C*, generalized linear model [GLM], 5 dpf vs. 6 dpf difference estimate = –2.26e-05, 95% CI = −3.84e-05 to −6.68e-06, z ratio = –3.357, p=0.002; 6 dpf vs. 7 dpf difference estimate = –9.73e-06, 95% CI = −2.25e-05 to 3.00e-06, z ratio = –1.806, p=0.1821; 7 dpf vs. 8 dpf difference estimate = –1.48e-06, 95% CI = −1.35e-05 to 1.05e-05, z ratio = –0.291, p=0.98). *gjd2b^-/-* PNs also showed a similar pattern of branch length growth, with an increase in TDBL from 5 to 6 dpf, followed by no significant increase until 8 dpf (*Figure 6C*). Post-hoc analysis revealed that there was a significant difference in TDBL between WT and gjd2b^-/- on all 4 days, with the mutant group having consistently lower values (*Figure 6A–C*, GLM, WT vs. mutant difference estimate = 8.17e-06, z ratio = 1.974, p=0.048). PN somata in WT and gjd2b^-/- larvae were comparable in diameter (*Figure 6—figure supplement 1A*, linear model, WT vs. mutant difference estimate = 0.3227, t = 1.935, p=0.055). These analyses reveal that loss of Gjd2b leads to a stunted dendritic arbor having fewer

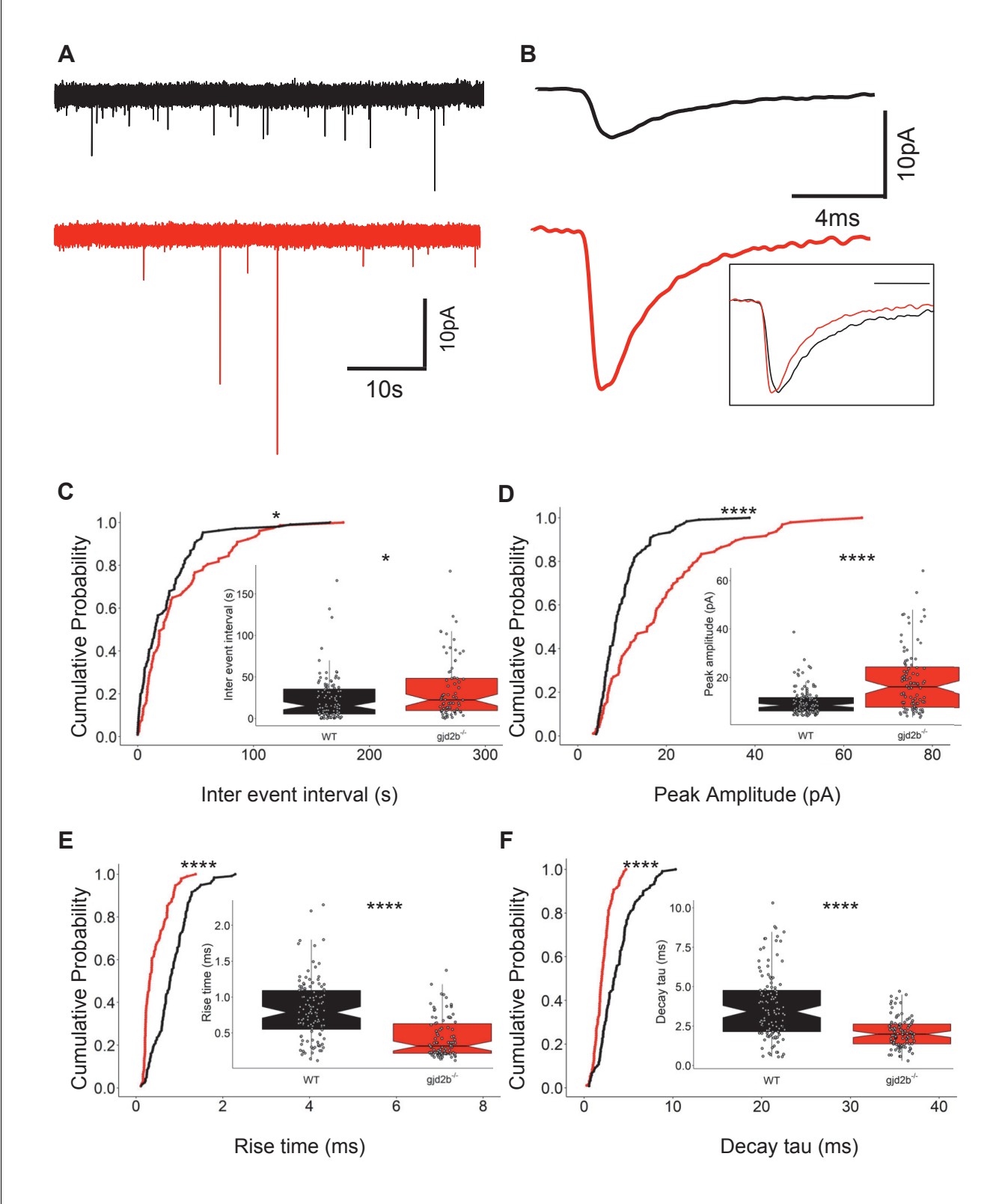

**Figure 3.** Knocking out Gjd2b results in decrease of glutamatergic synaptic number. (**A**) Representative miniature excitatory postsynaptic current (mEPSC) recordings from Purkinje neurons (PNs) of wild-type (black trace) and gjd2b$^{-/-}$ (red) larvae. (**B**) Average mEPSC shown on expanded time base recorded from wild-type (black) and gjd2b$^{-/-}$ (red) larvae. Neurons were held at −65 mV. Inset: scaled mEPSC to show faster rise time and decay time of mEPSCs in mutants. (**C–F**) Cumulative probability histograms and boxplots reveal increased inter-event intervals (**C**), increased peak

*Figure 3 continued on next page*

*Figure 3 continued*

amplitudes (**D**), decreased 10–90% rise times (**E**), and decreased decay time constants (**F**) of mEPSCs in gjd2b⁻/⁻ larvae (red lines) compared to wild type (black lines). N = 8 cells in wild type and 10 cells in gjd2b⁻/⁻ larvae. *p<0.05; ****p<0.0001; Mann–Whitney U test. Data used for quantitative analyses are available in *Figure 3—source data 1*. See also *Figure 3—figure supplement 1*.

The online version of this article includes the following source data and figure supplement(s) for figure 3:

**Source data 1.** Mutant mEPSC and Rin data.

**Figure supplement 1.** Input resistance of wild-type and mutant Purkinje neurons.

synapses. Since the arbor is stunted in mutants, the synapses that are present on the arbor are electrically closer to the soma compared to WT. This may in turn lead to the increased amplitudes and faster kinetics of mEPSCs recorded from the soma. These results are also consistent with the increased input resistance of mutant neurons compared to WT, seen only under conditions of better dendritic space clamp.

We also quantified the total dendritic branch numbers (TDBNs) of WT and mutant PNs at all 4 days. WT neurons showed a significant increase in branch numbers from 6 dpf to 7 dpf, but not at any other stages (*Figure 6—figure supplement 1C*, GLM with Poisson error distribution, 5 dpf vs. 6 dpf ratio = 1.132, z ratio = 2.175, p=0.0797; 6 dpf vs. 7 dpf ratio = 1.166, z ratio = 2.862, p=0.012; 7 dpf vs. 8 dpf ratio = 0.949, z ratio = –0.985, p=0.65). *gjd2b⁻/⁻* PNs also showed a similar pattern of branch number growth, with an increase in TDBN from 6 to 7 dpf, followed by no significant increase until 8 dpf (*Figure 6—figure supplement 1C*). Post-hoc analysis revealed that there was a significant difference in TDBN between WT and *gjd2b⁻/⁻* on all 4 days, with the mutant group having consistently higher values (GLM with Poisson error distribution, WT vs. mutant ratio = 1.16, z ratio = 3.708, p=0.0002). Despite these differences, WT and mutant neurons did not show any difference in the branching order at 7 dpf, as analyzed by Sholl analysis (*Figure 6—figure supplement 1B*, linear mixed model, WT vs. mutant difference estimate = –0.003208, z = –0.115, p=0.91).

To understand the dynamics that result in stunted and branched arbors, we imaged PNs every hour for 10 hr at 6 dpf. Mutant neurons had a significantly reduced growth rate compared to WT PNs (*Figure 6D*, linear model, WT vs. mutant difference estimate = 0.025462, t = 2.777, p=0.006). When dissected further into branch elongations and retractions, mutant neurons had a significantly lower rate of branch elongations than WT neurons (*Figure 6E*, linear model, WT vs. mutant difference estimate = 0.015030, t = 2.307, p=0.022), but their rates of branch length retractions were similar (*Figure 6F*, WT vs. mutant difference estimate = 0.009621, t = 1.33, p=0.185). This suggests that Gjd2b regulates dendritic growth by promoting branch elongations and is unlikely to be involved in regulating branch retractions.

## Functional Gjd2b in PNs alone is sufficient to rescue dendritic growth deficits

We next wished to determine which neurons are electrically coupled to PNs. Electroporation of single PNs with a combination of a high molecular weight dye (tetramethyl rhodamine dextran) and a low molecular weight tracer (neurobiotin/serotonin) failed to reveal any dye-coupled cells. However, when non-PNs were electroporated with neurobiotin or serotonin, one or two PNs along with several non-PNs were detected (*Figure 7—figure supplement 1*, *Supplementary file 1*), indicating that PNs are likely to be coupled to other cerebellar cell types via rectifying junctions. To determine if the observed dendritic growth deficits in *gjd2b⁻/⁻* mutant PNs are due to lack of Gjd2b in PNs specifically, we introduced Gjd2b tagged to mCherry into single PNs in *gjd2b⁻/⁻* fish (*Figure 7A*). TDBL in mutant PNs expressing Gjd2b were significantly larger than mutant PNs and were rescued to WT levels (*Figure 7B*, ANOVA post-hoc Tukey HSD, difference estimate = –74.6, t-ratio = –5.336, p<0.0001). This suggests that the presence of Gjd2b in single PNs in otherwise Gjd2b null larvae is sufficient to guide dendritic arbor elongation. Heterotypic gap junctional channels formed by Gjd2b on PN membranes and a different connexin isoform in the coupled cell could mediate this process.

To test if functional Gjd2b-mediated gap junctions are required for dendritic elaboration, we generated a construct coding for an N-terminal deleted version of Gjd2b (Gjd2bᐞ⁵⁻²¹) as the N-terminus of connexins has been shown to be required for channel function but not for assembly into gap junctional plaques (*Kyle et al., 2008*). Expression of Gjd2bᐞ⁵⁻²¹ in PNs of gjd2b⁻/⁻ larvae resulted in

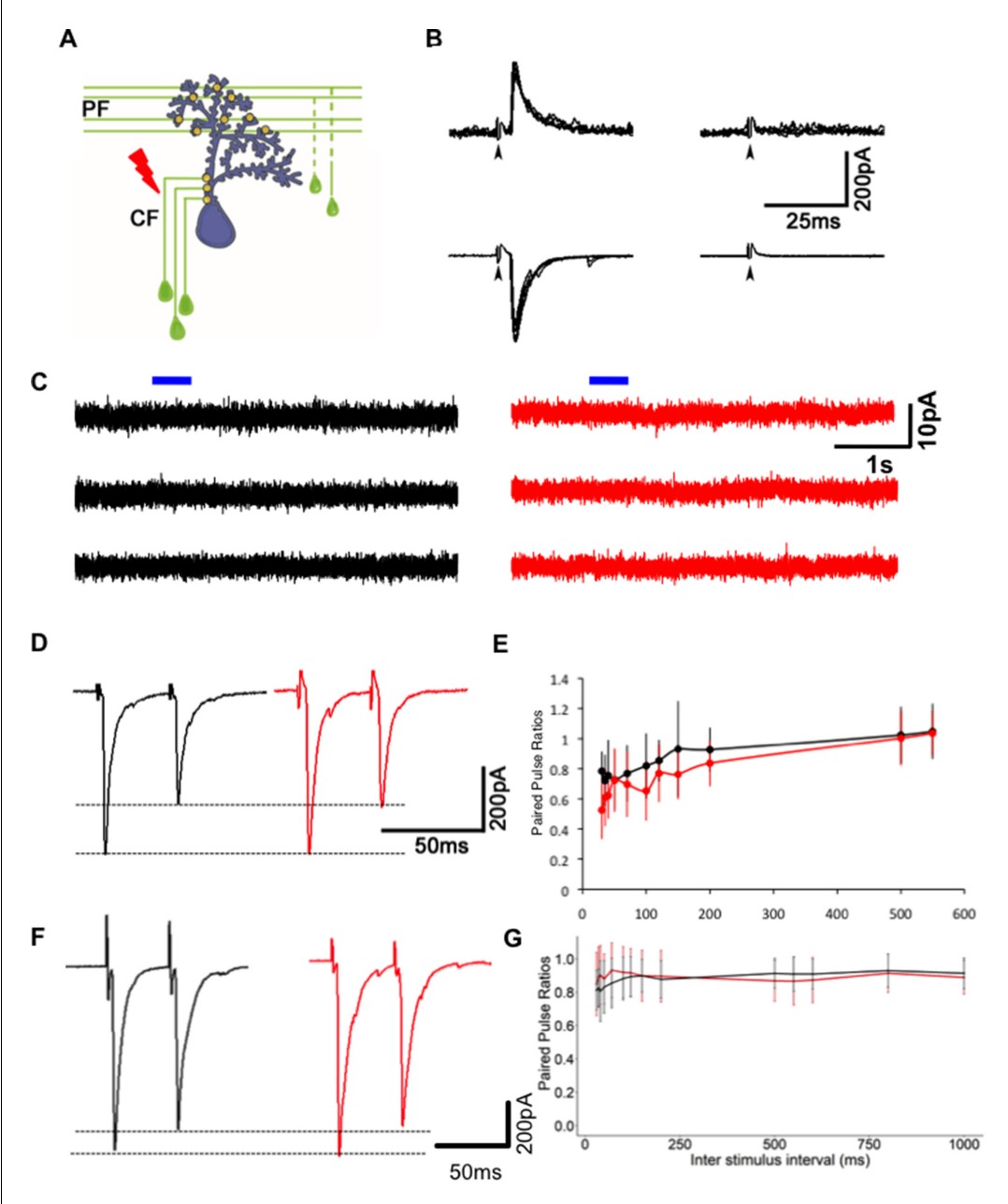

**Figure 4.** Reduction in miniature excitatory postsynaptic current (mEPSC) frequency in Gjd2b-KD and KO animals is not due to silent synapses or change in probability of transmitter release. (**A**) Schematic of experimental setup for stimulating climbing fibers (CFs) while recording EPSCs in Purkinje neurons (PNs; blue). (**B**) EPSCs recorded at a hyperpolarized holding potential (bottom row traces) and at a depolarized holding potential (top traces) in normal saline (left side traces) and in saline containing the AMPAR blocker CNQX (right-side traces). No EPSCs were detected in the presence of CNQX at −65 or +60 mV at 7 days post fertilization (dpf) (N = 4 cells) or at 19 dpf (N = 2 cells). Data from 19 dpf shown above. (**C**) Voltage clamp recordings from wild-type (WT) and mutant PNs at 7 dpf at a holding potential of −20 mV show no detectable response to brief pulses (blue bars) of N-methyl-D-aspartate (NMDA). Recordings were done in the presence of 1 μM tetrodotoxin (TTX). (**D**) Paired pulse depression of EPSCs in PNs of control (black) and Gjd2b-MO-injected (red) larvae. (**E**) Paired pulse ratios were not significantly different between control and Gjd2b-MO-injected larvae at any of the interstimulus intervals (ISIs) tested (mean ± SD; N = 5 cells from five larvae each in control and Gjd2b-MO groups; two-way repeated-measures ANOVA, p=0.081 for groups [control, Gjd2b-MO] and p<0.001 for ISIs). (**F**) Paired pulse depression of EPSCs in PNs of WT (black) and gjd2b−/− (red) larvae. (**G**) Paired pulse ratios were not significantly different between WT and mutant larvae. Mean ± SD; two-way repeated-measures ANOVA, p=0.17 for groups (control, Gjd2b-MO) and p<0.001 for ISI. Data used for quantitative analyses are available in *Figure 4—source data 1*. The online version of this article includes the following source data for figure 4:

**Source data 1.** Morphant and mutant PPR data.

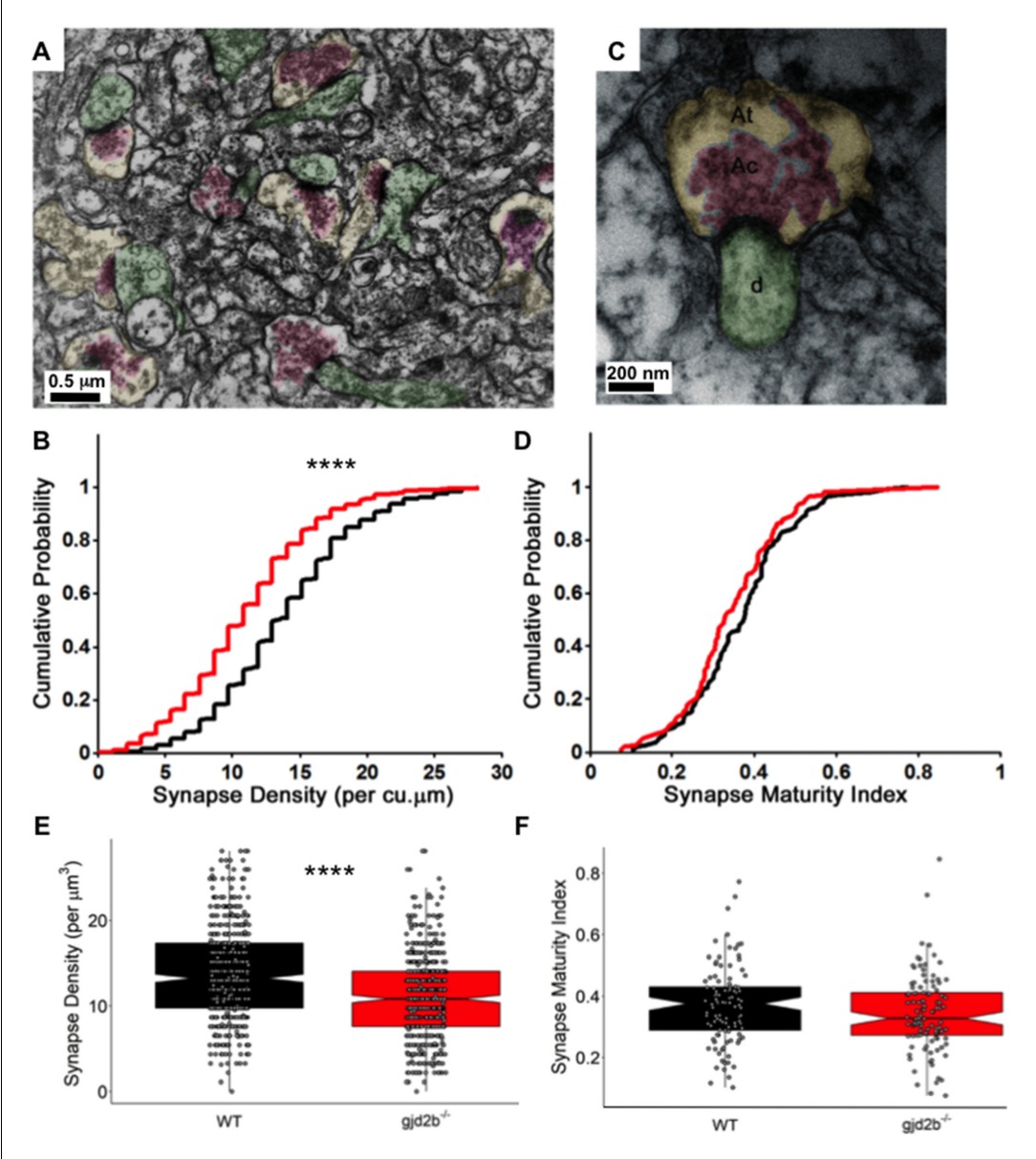

**Figure 5.** Knocking out Gjd2b leads to reduction in synaptic density in the cerebellar molecular layer. (A) Transmission electron micrograph illustrating synapses identified using clustered vesicles (pink areas) in presynaptic terminals (yellow areas) apposed to postsynaptic density in dendritic profiles (green areas). (B) Cumulative probability plot and (E) boxplot, showing distribution of synapse density per cubic micrometer in wild type (black) and gjd2b[-/-] larvae (red). 637 micrographs from three wild-type larvae and 550 micrographs from three mutant larvae were analyzed. ****p<0.0001; Mann–Whitney U test. (C) Transmission electron micrograph at high magnification used for quantification of synapse maturation index. The area occupied by clustered vesicles (pink, Ac) was divided by the total area of the presynaptic terminal (yellow, At) to obtain the maturation index. (D) Cumulative probability plot and (F) boxplot, showing the distribution of synapse maturation indices in wild type (black) and gjd2b[-/-] larvae. 106 micrographs from three larvae each in wild-type and mutant groups were analyzed. p=0.12, Mann–Whitney U test. See also *Supplementary file 1*. Data used for quantitative analyses are available in *Figure 5—source data 1*.

The online version of this article includes the following source data for figure 5:

**Source data 1.** Synapse density and maturity index data.

dendritic arbors that remained stunted and not significantly larger than mutant PNs lacking Gjd2b (*Figure 7B*, ANOVA post-hoc Tukey HSD, difference estimate = –18.5, t-ratio = –1.506, p=0.4370). These results point to the need for functional gap junctional channels in regulating PN dendritic elaboration.

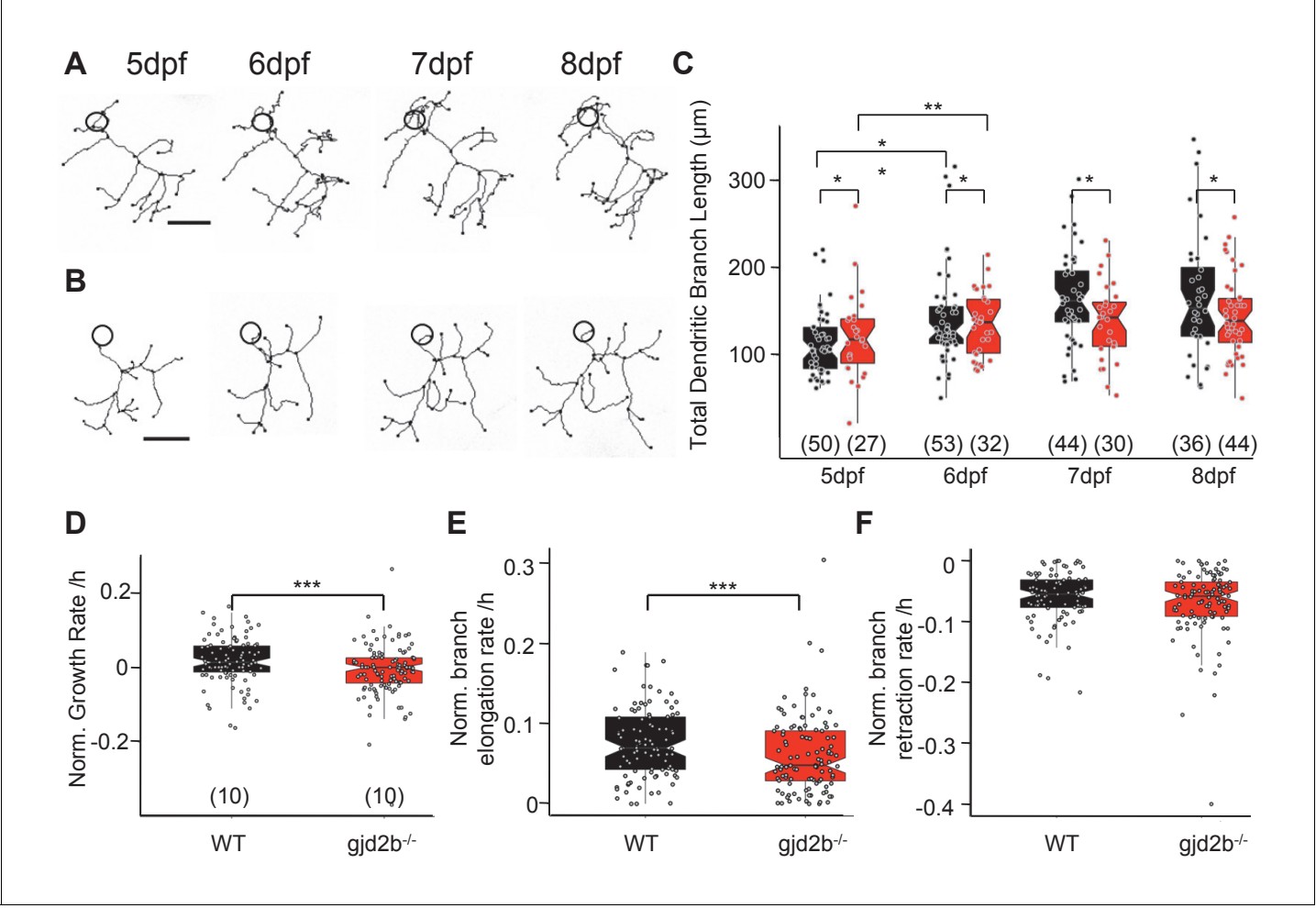

**Figure 6.** Dendritic arbor growth of gjd2b[-/-]Purkinje neurons (PNs) is impaired. (A) Representative traces of a wild-type (WT) PN from 5 to 8 days post fertilization (dpf). (B) Representative traces of a gjd2b[-/-] mutant PN from 5 to 8 dpf; scale bar is 10 μm; circle represents the position of the soma and is not to scale. (C) Total dendritic branch length (TDBL) of WT (black) and gjd2b[-/-] (red) PNs from 5 to 8 dpf. WT and gjd2b[-/-] neurons show significant growth from 5 to 6 dpf (p=0.0021). WT neurons show significantly higher TDBL values at all days (p=0.048). Statistical comparison was done using a general linear model with an inverse Gaussian error distribution. Post-hoc comparisons were done with the *emmeans* package in R. (D) Average rate of normalized hourly net branch growth in WT and gjd2b[-/-] PNs at 6 dpf. Mutant neurons show a significantly reduced rate over 10 hr of observation (p=0.005). (E) Rate of branch elongation in WT and mutant PNs at 6 dpf. Mutant PNs show a significantly reduced rate of branch length elongations (p=0.022). (F) Rate of branch length retractions in WT and mutant PNs at 6 dpf. No significant difference is observed between the two groups (p=0.185). Statistical comparisons in (D–F) were done using general linear models with Gaussian error distribution. Ns (number of neurons sampled) are indicated in parentheses in (C–E). See also *Figure 6—figure supplement 1*. Data used for quantitative analyses are available in *Figure 6—source data 1*. The online version of this article includes the following source data and figure supplement(s) for figure 6:

**Source data 1.** Daily and hourly imaging analysis.

**Figure supplement 1.** Additional morphometric comparison of Purkinje neurons (PNs) from wild-type (WT) and gjd2b[-/-] larvae.

## Presence of Gjd2b puncta on branches promotes their elongation

Next we wished to determine if the presence of Gjd2b puncta is sufficient to promote dendritic arbor growth at the level of single branches. We overexpressed full-length Gjd2b in PNs of WT larvae in a mosaic fashion. We imaged PNs at 5 dpf at 5 min intervals and observed the behavior of single dendritic branches. Dendritic branches having at least one Gjd2b punctum elongated more in length during the 5 min observation windows compared to branches that did not possess any puncta (Mann–Whitney test, U = 3779.5, p=0.022). Presence of Gjd2b puncta did not affect branch retraction lengths during the same window (*Figure 7C*, Mann–Whitney test, U = 4048, p=0.098). These results are consistent with the loss of Gjd2b affecting dendritic elongation but not retraction

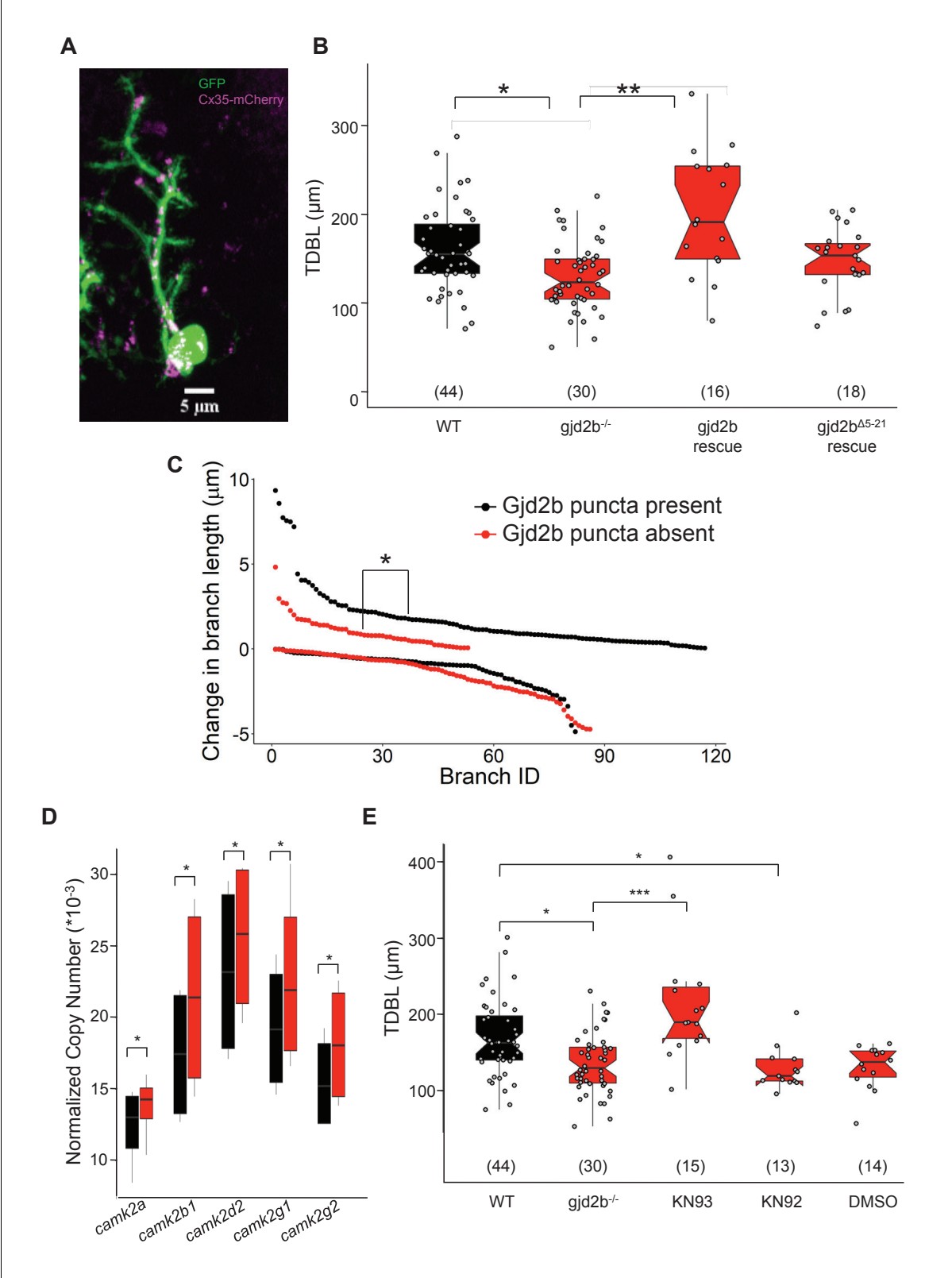

**Figure 7.** Expressing Gjd2b in Purkinje neurons (PNs) alone is sufficient to rescue dendritic growth deficits. (**A**) Representative image of a zebrafish PN expressing cytoplasmic GFP (green) and Gjd2b tagged with mCherry (magenta) at 7 days post fertilization (dpf); scale bar is 5 µm. (**B**) Total dendritic branch length (TDBL) of wild type (WT), gjd2b$^{-/-}$, gjd2b-rescue, and gjd2b$^{\Delta 5-21}$ rescue (pore dead variant) PNs at 7 dpf. TDBL of gjd2b-rescue neurons is significantly increased from that of mutant PNs. TDBL of gjd2b$^{\Delta 5-21}$ rescue PNs is not significantly different from that of gjd2b$^{-/-}$ PNs. (**C**) Change in the

*Figure 7 continued on next page*

*Figure 7 continued*

lengths of WT PN dendritic branches with and without Gjd2b-mCherry expression. Elongation of branches with Gjd2b is significantly more than branches without (p<0.05). Retraction of branches is similar with and without Gjd2b-mCherry puncta (Mann–Whitney U test; N = 7 neurons). (D) Copy number of CaMKII mRNA in WT and gjd2b$^{-/-}$ larvae. The normalized copy number of CaMKII in the mutant group is significantly higher than the WT group (p<0.05). (E) TDBL of WT, untreated gjd2b$^{-/-}$, 1 µM KN-93 treated, 1 µM KN-92 treated, and 1% DMSO-treated gjd2b$^{-/-}$ PNs at 7 dpf. TDBL of only KN-93-treated gjd2b$^{-/-}$ PNs are rescued to WT levels (WT vs. KN-93, p=0.23), whereas all other groups are significantly different and lower than WT (Kruskal–Wallis, post-hoc comparison with Mann–Whitney test). Ns (number of neurons sampled) are indicated in parentheses in (B) and (D). See also *Figure 7—figure supplement 1* and *Supplementary file 2*. Data used for quantitative analyses are available in *Figure 7—source data 1*.

The online version of this article includes the following source data and figure supplement(s) for figure 7:

**Source data 1.** Rescue and Branch dynamics data.

**Figure supplement 1.** Purkinje neurons (PNs) are dye coupled to other cerebellar cell types via putative rectifying junctions.

(*Figure 6E, F*). In sum, these results show that functional Gjd2b promotes the elongation of dendritic branches on which it is present.

## CaMKII inhibition can also rescue dendritic arbor growth in *gjd2b$^{-/-}$* larvae

To further understand how Gjd2b-mediated gap junctions regulate synaptogenesis and dendritic growth, we focused on cytoplasmic binding partners of Gjd2b that are also known to play a significant role in these processes. The calcium and calmodulin-dependent kinase II (CaMKII) has been shown to associate with Cx35/36-mediated gap junctions and modulate their function (*Pereda et al., 1998*; *Alev et al., 2008*; *Flores et al., 2010*; *Tetenborg et al., 2017*). CaMKII is localized in dendrites and in spines and is a critical regulator of dendritic development (*Wu and Cline, 1998*; *Zou and Cline, 1999*; *Wayman et al., 2008*). We first asked whether expression levels of the various isoforms of CaMKII are altered in *gjd2b$^{-/-}$* larvae compared to WT. To our surprise, we observed increased copy numbers for all isoforms of *camk2* we tested in *gjd2b$^{-/-}$* compared to WT (*Figure 7D*). CaMKII has previously been shown to reduce dendritic dynamics and stabilize branches (*Wu and Cline, 1998*; *Zou and Cline, 1999*). To test if the stunted arbors observed in *gjd2b$^{-/-}$* PNs were due to premature stabilization of dendritic arbors, mediated by increased CaMKII activity, we inhibited CaMKII in *gjd2b$^{-/-}$* larvae using the drug KN-93 in the embryo medium. Incubation in KN-93 was able to rescue dendritic arbor lengths of PNs in *gjd2b$^{-/-}$* larvae to WT levels (*Figure 7E*). No changes in dendritic arbor lengths were observed in larvae incubated in DMSO (vehicle control) or in KN-92, an inactive analog of KN-93 (*Figure 7E*). Taken together, these results show that conduction via Gjd2b-mediated electrical synapses is critical for proper glutamatergic synaptogenesis in PNs and that this process affects dendritic arbor growth.

## Discussion

### Gap junctions in PNs

Gap junction proteins are widely expressed in the cerebellum of developing and adult vertebrates. Cx36, the mammalian homolog of Gjd2b, has been shown to be expressed in the rodent cerebellar cortex in the molecular layer and granule cell layer (*Belluardo et al., 2000*; *Degen et al., 2004*; *Nagy and Rash, 2017*), and Cx36 puncta were observed localized to PNs (*Alcami and Marty, 2013*), although the mRNA was not found in PNs (*Belluardo et al., 2000*). In larval zebrafish, cell type-specific transcriptome analysis in larval zebrafish revealed gjd2b expression in PNs, granule cells, eurydendroid cells, and Bergmann glial cells (*Takeuchi et al., 2017*). We found Gjd2b puncta localized to PN cell membrane, but when dye was injected into PNs, we failed to observe any dye-coupled cells. This may be attributable to the low conductance of zebrafish Gjd2b channels, which have a unitary conductance of around 24 pS (*Valiunas et al., 2004*). This is also true of Cx36, which have very small unitary conductances of 10–15 pS (*Srinivas et al., 1999*). Cx36 has been shown to not support dye coupling (*Teubner et al., 2000*; *Quesada et al., 2003*). Further, when these same cells were made to overexpress Cx32, which has a larger conductance compared to Cx36, dye coupling could be observed (*Quesada et al., 2003*). Electrical coupling in the absence of dye coupling has been widely observed (*Ransom and Kettenmann, 1990*; *Meda et al., 1991*; *Pérez-*

*Armendariz et al., 1991*; *Moser, 1998*; *Teubner et al., 2000*; *Quesada et al., 2003*). From these studies, it appears that PNs in fish and mammals are electrically coupled to other neurons, likely mediated by Cx36 in mammals and Cx35 in zebrafish. We suggest that these gap junctions on PNs could serve important developmental functions in all vertebrates.

## Regulation of chemical synapse formation by electrical synapses

Our results indicate that Gjd2b-mediated functional electrical synapses are important regulators of glutamatergic synapse formation and dendritic elaboration. These results are in agreement with earlier studies on innexin-mediated gap junctions in invertebrates. Knockdown of the innexin *inx1* in leech resulted in loss of electrical coupling between identified neurons at embryonic stages and decreased chemical synaptic strength between the same neurons at much later stages (*Todd et al., 2010*). In the neocortex of mice, sister neurons born from the same radial glia make transient electrical synapses, which are required for the formation of excitatory connections between them (*Yu et al., 2009*; *Yu et al., 2012*). Mice lacking Cx36 exhibit reduced synaptic connectivity between mitral cells in the olfactory bulb (*Maher et al., 2009*). Interestingly, though Cx36 is the predominant neural connexin, deficits in glutamatergic synapse formation have hitherto not been reported from other regions of the CNS in the Cx36$^{-/-}$ mouse, to the best of our knowledge. However, an increase in the number of inhibitory synapses was observed in Cx36$^{-/-}$ mice in thalamocortical relay neurons with a concomitant decrease in their dendritic complexity (*Zolnik and Connors, 2016*). Using morpholino-mediated knockdown and knockout approaches, we show that glutamatergic synapse number is decreased significantly when Gjd2b/Cx35b is perturbed. The decrease in synapse number was seen using both structural (transmission electron microscopy) and functional (electrophysiology) assays. In addition, there was a concomitant decrease in dendritic arbor size. The amplitude and kinetics of mEPSCs of mutant PNs followed a trend that was consistent with smaller dendritic arbors. However, alternate explanations such as changes in receptor numbers and subunit types are also possible. We present the results, interpretation, and caveats of all the experiments in this work in tabular form in *Supplementary file 4*. The collective evidence presented in this paper shows that Gjd2b-mediated functional gap junctions are important regulators of chemical synapse formation and dendritic elaboration in PNs.

## Dendritic development of PNs

PNs exhibit one of the most elaborate and beautiful dendritic arbors known, and a number of studies have examined factors that determine arbor structure in PNs. In rodents, PN dendritic development occurs over a period of 1 month after birth and involves multiple steps. Rodent PNs undergo morphological changes soon after they reach the Purkinje cell layer. They retract their simple fusiform dendrites and then acquire a stellate morphology with multiple dendrites sprouting from their somata (*Kapfhammer, 2004*; *Sotelo and Dusart, 2009*; *Tanaka, 2009*). This has also been observed in zebrafish PNs (*Tanabe et al., 2010* and Sitaraman and Thirumalai, unpublished observations). Later, one of these becomes the primary dendrite and the others are retracted. This process involves the localization of Golgi organelles at the base of the primary dendrite and is mediated by an atypical PKC (*Tanabe et al., 2010*). These early steps occur roughly before 4 dpf in zebrafish and before P10 in rodents. The early remodeling is mainly dependent on intrinsic factors, and the overall architecture of PN dendrites is maintained even in the absence of afferent input or activity. In the second postnatal week, the apical dendrite of rodent PNs is spiny and undergoes growth and branching to occupy the molecular layer in a planar manner. By P20 in mice and P30 in rats, PN dendrites have achieved their maximal length (*Kapfhammer, 2004*; *Sotelo and Dusart, 2009*). From our results, it appears that significant growth of PN dendritic arbors occurs in zebrafish between 5 and 8 dpf. At these stages, the neurons are spiny and receive parallel and CF inputs (*Sengupta and Thirumalai, 2015*). Both WT and mutant neurons show large variability in their TDBLs and numbers at all days of observation. Such variability could reflect distinct subtypes within the PN population or could be related to their time of birth. We also observed that at these stages the dendritic branches are dynamic and undergo elongations and retractions. In rodents, lack of afferent input during this phase results in abnormal orientation, reduced size, and lack of higher-order branches in PN dendritic arbors (*Altman and Anderson, 1972*; *Rakic and Sidman, 1973*). In both slice cultures and dissociated cell cultures, blockade of glutamatergic transmission reduces dendritic arbor size of PNs

(*Catania et al., 2001*; *Adcock et al., 2004*). We observed that dendritic arbors of gjd2b[-/-] PNs were smaller compared to WT even at 5 dpf and stayed smaller at least until 8 dpf. During these stages, they also received fewer glutamatergic mEPSCs, suggesting fewer synaptic contacts. It is likely that the reduced glutamatergic synapses in gjd2b[-/-] PNs lead to a stunted dendritic arbor via a synapto-trophic mechanism (*Haas et al., 2006*; *Cline and Haas, 2008*). It is also likely that Gjd2b directly affects dendritic growth by promoting branch elongations. In gjd2b[-/-] PNs, lack of Gjd2b puncta results in shorter branch elongations as demonstrated in *Figure 7C*, and therefore a stunted arbor. A ping-pong mechanism, whereby the smaller dendritic arbor of gjd2b[-/-] PNs restricts the number of functional synapses that can be formed and the reduced afferent input in turn restricts further dendritic growth, could underlie this process.

We could rescue dendritic arbor growth deficits in Gjd2b mutant zebrafish by expressing full-length Gjd2b in single PNs. In addition, we found that expressing an N-terminal deleted, pore-dead version of Gjd2b could not rescue the dendritic growth deficit. These results suggest that conduction of signaling molecules through Gjd2b-mediated gap junctions regulates dendritic arborization. Further, in WT PNs, Gjd2b puncta facilitate dendritic branch elongation while not affecting branch retractions (*Figure 7C*). These data together lead us to a model whereby Gjd2b-containing gap junctions conduct signals that promote the elongation of dendritic branches and the formation of glutamatergic synapses locally.

### Role of CaMKII in gap junction-mediated PN development

Signaling via Gjd2b-containing gap junctions could lead to long-lasting global changes such as the increased expression levels of α, β1, δ2, γ1, and γ2 isoforms of CaMKII that we observed. Further experiments are required to verify whether this increase in expression level of CaMKII isoforms translates to increased enzymatic activity. Nevertheless, when CaMKII levels and/or activity increase, dendrites are stabilized at their mature lengths. Lack of gap junctional signaling leads to premature stabilization of dendritic branches leading to stunted growth. In *Xenopus* tectum, immature neurons with simple arbors and low levels of CaMKII continue to grow while mature neurons have high levels of CaMKII and their dendritic structure is more or less stable. Expression of constitutively active CaMKII in tectal neurons causes them to grow slower and have relatively less dynamic arbors. In addition, inhibition of CaMKII in mature neurons causes them to grow at a higher rate (*Wu and Cline, 1998*). More recently, a human CAMK2A mutation, isolated from an ASD proband, was shown to cause increased dendrite arborization, when the mutant CAMK2A was introduced into cultured mouse hippocampal neurons (*Stephenson et al., 2017*). Our results are consistent with these earlier findings and suggest a stabilizing role for CaMKII in PN dendritic arbor elaboration. The mechanisms by which signaling via Gjd2b gap junctions regulate CaMKII levels will have to be investigated in future experiments.

## Materials and methods

### Key resources table

| Reagent type (species) or resource | Designation | Source or reference | Identifiers | Additional information |
|---|---|---|---|---|
| Genetic reagent (*Danio rerio*) | gjd2b[ncb215] | This paper | RRID:ZDB-ALT-201215-7 | Gjd2b null zebrafish allele ncb215 |
| Recombinant DNA reagent | aldoca:gap43-Venus | Prof. Masahiko Hibi, Nagoya University, Japan *Tanabe et al., 2010* | | Microinjected in single-cell zebrafish embryos |
| Recombinant DNA reagent | Arch:TagRFP-T:PC:GCAMP5G | Dr. Hideaki Matsui, Niigata University, Japan *Matsui et al., 2014* | | Microinjected in single-cell zebrafish embryos |
| Recombinant DNA reagent | Ca8-cfos:GFP | This paper | PN enhancer to drive GFP expression | Microinjected in single-cell zebrafish embryos |

*Continued on next page*

*Continued*

| Reagent type (species) or resource | Designation | Source or reference | Identifiers | Additional information |
|---|---|---|---|---|
| Recombinant DNA reagent | Ca8-cfos:Gjd2b-mCherry | This paper | PN enhancer to drive Gjd2b-mCherry expression | Microinjected in single-cell zebrafish embryos |
| Recombinant DNA reagent | Ca8-cfos:Gjd2b$^{⊗5-21}$-mCherry | This paper | PN enhancer to drive Gjd2b deletion mutant | Microinjected in single-cell zebrafish embryos |
| Recombinant DNA reagent | pTNT | Promega Corp, Madison, WI | | Vector for in vitro transcription |
| Transfected construct (*Danio rerio*) | gjd2b-TALEN-1 | This paper | TALEN construct | GAACAGCCATGGGGGAATGGA |
| Transfected construct (*Danio rerio*) | gjd2b-TALEN-2 | This paper | TALEN construct | GCTGTTGGACAGCCGCCTCCA |
| Commercial assay or kit | T7-mMessage mMachine | Life Technologies | | For generating TALEN mRNAs |
| Antibody | Anti-Cx35/36 (Mouse) | Millipore | Cat# MAB3045 | 1:250 |
| Antibody | Anti-Parvalbumin-7 (mouse) | Millipore | MAB1572 | 1:1000 |
| Antibody | Donkey anti-mouse Alexa Fluor 488 | Invitrogen | A21202 | 1:500 |
| Antibody | Goat anti-rabbit Alexa Fluor 488 | Invitrogen | A21052 | 1:1000 |
| Other | Prolong gold antifade reagent | Molecular Probes | Catalog #P10144 | Mounting reagent |
| Sequence-based reagent | Gjd2b-MO | Gene Tools | Splice block morpholino | 5'ACAACACTTTTTCC CCTCACCTCCC3' |
| Sequence-based reagent | CTRL | Gene Tools | Control morpholino | 5'ACTAGACTTATTCC CGTGACCTCCC3' |
| Sequence-based reagent | Gjd2b forward | This paper | PCR primers | 5'GATCGGTACCTCCGA ATGAACAGCCAT3' |
| Sequence-based reagent | Gjd2b reverse | This paper | PCR primers | 5'TAGCGCTAGCAACGTA GGCAGAGTCACTGG3' |
| Sequence-based reagent | Gjd2b$^{⊗5-21}$ forward | This paper | PCR primers | 5'ATTGCCATGGGGGAATGGA TTGGGAGGATCCTGCTAAC3' |
| Sequence-based reagent | Gjd2b$^{⊗5-21}$ reverse | This paper | PCR primers | 5'TAGCGCTAGCAACGTA GGCAGAGTCACTGG3' |
| Chemical compound, drug | MS-222 (Tricaine) | Sigma-Aldrich | CAS #886862 | |
| Chemical compound, drug | Paraformaldehyde | Alfa Aesar | Catalog #47392-9M | |
| Chemical compound, drug | Tetrodotoxin | Hello Bio | Catalog #1069 | 1 μM |
| Chemical compound, drug | NMDA | Tocris | Catalog #0114 | 10 mM |
| Software | R statistical software | R core team https://www.r-project.org | RRID:SCR_001905 | |
| Software | MATLAB | MathWorks https://www.mathworks.com/products/matlab.html | RRID:SCR_001622 | |
| Software | Clampfit 10.2 | Molecular Devices | RRID:SCR_011323 | |
| Software | Fiji | NIH | http://fiji.sc RRID:SCR_002285 | |

*Continued on next page*

*Continued*

| Reagent type (species) or resource | Designation | Source or reference | Identifiers | Additional information |
|---|---|---|---|---|
| Software | Quantsoft | Bio-Rad | Version 1.7.4 | |
| Other | QX200 AutoDG Droplet Digital PCR system | Bio-Rad | 1864100 | Equipment |
| Other | LSM 780 confocal microscope | Zeiss | RRID:SCR_020922 | Equipment |
| Other | SP5 point scanning confocal microscope | Leica | | Equipment |
| Other | FV3000 confocal microscope | Olympus | RRID:SCR_017015 | Equipment |
| Other | Ultramicrotome | Power Tome-PC | | Equipment |
| Other | Diamond knife | Electron Microscopy Sciences | | |
| Other | Formavar/Carbon 2 × 1 mm copper or nickel slot grids | Electron Microscopy Sciences | | |
| Other | TECNAI T12 G$^2$ Spirit BioTWIN transmission electron microscope | FEI Company | | Equipment |
| Other | Borosilicate glass capillaries | Warner Instruments | | OD: 1.5 mm; ID: 0.86 mm |
| Other | Flaming-Brown P-97 pipette puller | Sutter Instruments | RRID:SCR_020540 | Equipment |
| Other | Bipolar electrode | FHC, Bowdoin | | |
| Other | Multiclamp 700b amplifier | Molecular Devices | RRID:SCR_018455 | Equipment |
| Other | Digidata 1440A digitizer | Molecular Devices | | Equipment |

## Zebrafish and animal husbandry

All experiments were performed using Indian WT zebrafish. Institutional Animal Ethics and Biosafety committee approvals were obtained for all procedures adopted in this study. Larvae and adults were reared using standard procedures (*Westerfield, 2000*).

## Generation of transient transgenic larvae

To label single PNs for some of the experiments, single-celled embryos were microinjected with one of the following constructs along with Tol2 transposase mRNA (*Urasaki et al., 2006*): aldoca:gap43-Venus (*Tanabe et al., 2010*) (gift from Prof. Masahiko Hibi, Nagoya University, Japan); Ca8-cfos: RFP (*Matsui et al., 2014*) (gift from Dr. Hideaki Matsui, Niigata University, Japan); Ca8-cfos:GFP; Ca8-cfos:Gjd2b-mCherry; Ca8-cfos:Gjd2b$^{\Delta 5-21}$-mCherry. Ca8-cfos: GFP was constructed by amplifying Ca8-cfos from the parent plasmid and ligating it with sequences coding for GFP. To construct the last two plasmids, full-length Gjd2b and Gjd2b$^{\Delta 5-21}$ were first amplified from total cDNA using the following primers: Gjd2b forward: 5′GATCGGTACCTCCGAATGAACAGCCAT3′; Gjd2b reverse: 5′TAGCGCTAGCAACGTAGGCAGAGTCACTGG3′; Gjd2b$^{\Delta 5-21}$ forward: 5′ATTGCCATGGGGGGAA TGGATTGGGAGGATCCTGCTAAC3′; Gjd2b$^{\Delta 5-21}$ reverse: 5′TAGCGCTAGCAACGTAGGCAGAG TCACTGG3′. The amplified regions were digested using *NcoI* and *NheI* and ligated with Ca8-cfos on the 5′ end and mCherry at the 3′ end to generate the respective plasmids. Microinjected embryos were reared in embryo medium containing 0.003% of 1-phenyl-2-thiourea (PTU) for imaging experiments.

## Morpholino antisense oligonucleotide-mediated knockdown of Cx35

Morpholino antisense oligonucleotides (Gene Tools LLC) were designed to bind at the junction between exon 1 and intron 1 of the gjd2b mRNA to block proper splicing of the mRNA (Gjd2b-MO; *Figure 1—figure supplement 2A*). Control morpholinos (CTRL) were designed to incorporate mismatches at five positions within the gjd2b recognition sequence. The morpholino sequences were:

Gjd2b-MO: 5′ ACAACACTTTTTCCCCTCACCTCCC 3′
CTRL: 5′ ACTAGACTTATTCCCGTGACCTCCC 3′

Either Gjd2b-MO or CTRL were injected into single-celled zebrafish embryos at 0.05 pmoles per embryo.

## Generation of gjd2b$^{-/-}$ zebrafish

Transcription activator like effector nucleases (TALENs) recognizing nucleotide sequences near the start codon of gjd2b gene were used to generate the gjd2b mutant (*gjd2b$^{-/-}$*) lines of zebrafish used in this study (*Figure 2*). A pair of TALEN vector constructs were designed and assembled to generate gjd2b-TALEN-1 and gjd2b-TALEN-2, which bind the plus and minus strands of *gjd2b* gene (*Figure 2A*), respectively, by following published protocols (*Sanjana et al., 2012*). These TALEN sequences were later moved to a pTNT (Promega Corp, Madison, WI) vector. TALEN mRNAs that encode gjd2b-TALEN-1 and gjd2b-TALEN-2 proteins were synthesized in vitro from the above vectors using the T7-mMessage mMachine kit (Life Technologies, Thermo Fisher Scientific, USA) and micro-injected into one-cell stage WT zebrafish embryos at a concentration of 50 ng/μl of each mRNA. TALENs were designed such that the spacer region incorporated an *Xho1* restriction site enabling an easy screen for mutations in this locus. Up to 10 embryos were taken from every clutch of TALEN-injected embryos and screened for the presence of mutations using *Xho1* restriction digestion. Clutches of embryos that showed a high percentage of mutants were then grown up into adults. The TALEN-injected founder generation (F0) was raised in the facility and the adult fish were out-crossed with WT fish to get heterozygous F1 progeny. These were screened for germline transmission of mutations in the *gjd2b* locus. Whole embryos or adult fish tail clips were screened using *Xho1* restriction analysis of 800 bp DNA band around the TALEN-target site followed by sequencing of this region (*Figure 2B, C*). F1 heterozygous mutants (*gjd2b$^{+/-}$*) were inbred to obtain F2 WT siblings, heterozygotes, and homozygotes (*gjd2b$^{-/-}$*).

## Single-cell electroporation

5 dpf zebrafish larvae were embedded in 1.5% low gelling agarose (Sigma) in a dorsoventral position. The Purkinje cell layer was observed in these larvae under the 63× water immersion objective of a Nikon compound microscope. Patch pipettes (OD: 1.5 mm, ID: 0.86 mm) were pulled using borosilicate glass capillaries and a P-97 pipette puller (Sutter Instruments). A single pipette was backfilled with a mixture of tetramethylrhodamine dextran (TMR-dextran) and serotonin/neurobiotin and inserted through the skin of the larva. The pipette tip was positioned near a cell body in the Purkinje cell layer and 3–5 electric pulses (30 V, 30 ms) were administered, until the neuron was completely filled with the dye. The larva was then released from the agarose and fixed in 4% paraformaldehyde after 30 min and processed for visualizing the injected serotonin or neurobiotin.

## Whole-mount immunohistochemistry

Whole-mount immunofluorescence was performed as described in *Jabeen and Thirumalai, 2013*. Briefly, 5–7 dpf larvae were anesthetized in 0.01% chilled tricaine (Sigma-Aldrich) and then fixed overnight at 4°C in 4% paraformaldehyde (Alfa Aesar, Thermo Fisher Scientific, UK), followed by several washes in 0.1 M phosphate buffered saline (PBS) at room temperature. The eyes, jaws, and yolk sac were carefully dissected out and the skin covering the brain was peeled to expose the brain. Dissected larvae were kept overnight in 5% normal donkey serum and 0.5% Triton-X100 in 0.1M PBS (PBST) and at 4°C. Next, they were incubated in a mouse anti-Cx35/36 antibody (MAB3045, EMD Millipore, Merck, USA) at a dilution of 1:250 for labeling Gjd2b puncta or rabbit anti-serotonin antibody (Sigma-Aldrich) at a dilution of 1:500 for labeling the serotonin electroporated cells. In larvae in which PNs were stochastically labeled with EGFP, we used the chicken anti-EGFP antibody (Ab13970, Abcam, UK). Primary antibody treatments were done in blocking solutions for 48 hr. After

several washes in PBST, larvae were incubated overnight in donkey anti-mouse Alexa Fluor 488 antibody to label Gjd2b puncta or goat anti-mouse Alexa Fluor 633 to label PNs (A21202 or A11034 and A21052, Invitrogen, USA) at a dilution of 1:500 or 1:1000, respectively, in blocking solutions. Following this, the larvae were washed in 0.1 M PBS several times and then mounted between two coverslips using Prolong Gold antifade reagent (Molecular probes, Life Technologies, Thermo Fisher Scientific) and stored in the dark at $4^0$ C until imaging.

## Confocal imaging and image analysis

Images were acquired on these confocal laser scanning microscopes: Zeiss LSM 780 using 63× oil immersion objective, Olympus FV1000 using a 60× oil immersion objective and Olympus FV3000 confocal laser scanning microscope with a 60× oil immersion objective. Imaging parameters and conditions were the same for all larvae imaged in their respective groups. Dorsal-most regions of the cerebellum were imaged. Two images were taken from each animal, one in each hemisphere. Image analysis was done using Fiji (https://fiji.sc/; *Schindelin et al., 2012*). For Gjd2b puncta analysis, a median filter was applied to remove salt and pepper noise from images after a background subtraction (rolling ball method). Average intensity of the z-projections of image stacks was then measured and normalized with respect to the intensity of uninjected (*Figure 1—figure supplement 2*) or WT animals (*Figure 2*). These were then plotted using R statistical software (https://www.r-project.org).

## Transmission electron microscopy and image analysis

Zebrafish larvae at 7 dpf were fixed overnight in 4% paraformaldehyde and 2.5% glutaraldehyde prepared in EM buffer (70 mM sodium cacodylate and 1 mM $CaCl_2$, pH 7.4). Secondary fixation was done with 1% $OsO_4$ made in EM buffer for 90 min on ice. For additional contrast, samples were incubated in aqueous 2% uranyl acetate for 1 hr at room temperature. Samples were dehydrated serially in 30%, 50%, 70%, 90%, and finally in absolute ethanol for 15 min each. After two changes of ethanol (15′ each), samples were incubated in acetone with two changes for 10 min each. Samples were infiltrated with Epon812-Araldite resin mix and acetone in the ratio of 1:3, 1:1, 3:1 and in pure Epon812-Araldite mix for 2 hr, overnight, 2 hr and overnight, respectively. Samples were then embedded in the Epon812-Araldite resin mix for polymerization at 60°C for 48 hr. After polymerization, thick transverse sections were cut using a glass knife till the region of interest was obtained. Then, 60 nm ultrathin sections were cut from the anterior end of the cerebellum on an ultramicrotome (Power Tome-PC, RMC Boeckeler) by using 3.5 size Ultra 45° diamond knife (Electron Microscopy Sciences, USA). Sections were collected on Formavar/Carbon 2 × 1 mm copper or nickel slot grids (FCF 2010-Cu/Ni, Electron Microscopy Sciences) for imaging. Images were acquired on FEI TECNAI T12 $G^2$ Spirit Bio-TWIN transmission electron microscope. For counting synapses and measuring synapse maturation indices, images were taken at 30K and 50K magnification, respectively. Total volume of a single imaged micrograph was 0.924 cubic micrometers. Images were analyzed using Fiji. Analysis was done blind to the genotype.

## Electrophysiology

Whole-cell patch clamp recording from larval PNs was performed as described in *Sengupta and Thirumalai, 2015*. Briefly, 7 dpf larvae were anesthetized in 0.01% MS222 and then pinned onto a piece of Sylgard (Dow Corning) glued to a recording chamber. Then larvae were submerged in external saline (134 mM NaCl, 2.9 mM KCl, 1.2 mM $MgCl_2$, 10 mM HEPES, 10 mM glucose, 0.01 D-tubocurarine, 2.1 mM $CaCl_2$; pH 7.8; 290 mOsm). The cerebellum was exposed by peeling the skin from the top of the head. Cells were viewed using the 63× water immersion objective of a fixed stage compound microscope (Nikon Ni-E or Olympus BX61WI). Patch pipettes were pulled from borosilicate glass (OD: 1.5 mm; ID: 0.86 mm; Warner Instruments) on a Flaming-Brown P-97 pipette puller (Sutter Instruments), filled with pipette internal solution, and had a resistance of 10–12 MΩ. For mEPSC recordings, cesium gluconate pipette internal solution was used (115 mM CsOH, 115 mM gluconic acid, 15 mM CsCl, 2 mM NaCl, 10 mM HEPES, 10 mM EGTA, 4 mM Mg-ATP; pH 7.2; 290 mOsm). mEPSCs were recorded after bath application of TTX (1 μM), gabazine (10 μM), and APV (40 μM). Cells were held in voltage clamp at −65 mV for recording AMPAR-mediated currents. In a few cells, the holding potential was varied to measure reversal potential for the mEPSCs.

Pipettes also contained sulforhodamine dye (Sigma), and only those cells filled with the dye at the end of the recording and showing Venus expression were considered for further analysis. In addition, cells whose series resistance varied by more than 20% during the recording session or those that had input resistances lower than 1 GΩ were excluded.

For evoked synaptic current recording, potassium gluconate-based internal solution was used (115 mM K gluconate, 15 mM KCl, 2 mM $MgCl_2$, 10 mM HEPES, 10 mM EGTA, 4 mM Mg-ATP; pH: 7.2; 290 mOsm). Evoked synaptic currents were recorded by stimulating CFs with a bipolar electrode (FHC, Bowdoin, ME, USA). Stimulus strength was gradually increased until no failures occurred. PPR was measured by stimulating CFs at various ISIs ranging from 30 ms to 550 ms (in morphants) and 1000 ms (in mutants) and calculating the ratio of the peak amplitude of the second EPSC to the first.

For measuring NMDAR-mediated currents, all recordings were done with 1 µM TTX (Hello Bio catalog no. 1069) to block action potentials and with cesium gluconate internal in the recording pipette. Cells were held at a depolarized potential (−20 mV) to remove the $Mg^{2+}$ block from the NMDAR. 10 mM NMDA (Tocris catalog no. 0114) was released close to the soma that was being recorded for 500 ms with a Picospritzer. Three trials per cell were recorded with each trial lasting 12 s.

Signals were acquired using Multiclamp 700b amplifier and digitized with Digidata 1440A digitizer (Molecular Devices). Data analysis was done offline in Clampfit 10.2 (Molecular Devices), and statistical analysis was performed using in-built functions in MATLAB (The MathWorks, Natick, MA). mEPSCs were analyzed blind to the experimental group.

### In vivo time-lapse imaging and analysis

For daily imaging, 5 dpf larvae with sparse fluorescent protein labeling of PNs were anesthetized in 0.001% MS222 (Sigma) for 1 min. They were then mounted in 1.5% low gelling agarose (Sigma) on a custom-made confocal dish with their dorsal side towards the coverslip and covered completely with embryo medium. PNs in embedded larvae were imaged using a Leica SP5 point scanning microscope. Neurons were chosen based on their morphology, level of RFP/GFP expression and traceability. Imaging parameters were kept constant across samples. Post imaging, the larvae were released from agarose and allowed to recover in embryo medium for 24 hr. They were imaged again at 6, 7, and 8 dpf. For the hourly imaging, PNs in 6 dpf larvae were imaged at 1 hr intervals for 10 hr. The relationship between Gjd2b puncta and branch dynamics was investigated by imaging PNs in 5 dpf larvae at 5 min intervals for 30 min. For gjd2b and gjd2b$^{Δ5-21}$ rescue experiments, PNs with both GFP and punctate mCherry in 7 dpf larvae were imaged using an Olympus FV3000 confocal microscope. Neurons were traced using the Simple Neurite tracer plugin in Fiji and their TDBL and TDBN were calculated.

For analyzing the daily imaging data, TDBL and TDBN of the two genotypes, across days, were statistically compared using GLM (inverse Gaussian and Poisson, respectively). For post-hoc analysis, the *emmeans* package in R was used to compare the TDBL and TDBN of consecutive days within each genotype and that of same days across the two genotypes. Hourly growth rates, elongation rates, and retraction rates of WT and mutant PNs were compared using GLM with Gaussian error distribution. For analyzing the effects of gjd2b and gjd2b$^{Δ5-21}$ rescue, the TDBL and TDBN across groups were statistically compared using ANOVA and post-hoc Tukey HSD.

## Acknowledgements

We thank the following sources of funding support: Wellcome Trust-DBT India Alliance Intermediate and Senior fellowships (VT; 500040/Z/09/Z and IA/S/17/2/503297), Department of Biotechnology (VT; BT/PR4983/MED/30/790/2012), Science and Engineering Research Board, Department of Science and Technology (VT; EMR/2015/000595), Department of Atomic Energy (VT; 12R and D-TFR-5.04-0800), CSIR-UGC fellowship (SJ), NCBS-TIFR graduate student fellowship (SS, VA, SV, MJ, and SJ), and SERB Young Scientist Scheme (GY). We would also like to thank Prof. Masahiko Hibi for the aldoca construct and Dr. Hideaki Matsui for the Ca8 enhancer construct. Further thanks are also due to Dr. Igor Kondrychyn, Mr. Sriram Narayanan for technical assistance, and Mr. PT Jagadeesh for the maintenance of our fish lines. In addition, we would like to thank the NCBS-TIFR Genomics facility and the Central Imaging and Flow Facility for support.

## Additional information

### Competing interests

Vatsala Thirumalai: Reviewing editor, *eLife*. The other authors declare that no competing interests exist.

### Funding

| Funder | Grant reference number | Author |
|---|---|---|
| Wellcome Trust DBT India Alliance | 500040/Z/09/Z and IA/S/17/2/503297 | Vatsala Thirumalai |
| Department of Biotechnology, Ministry of Science and Technology, India | BT/PR4983/MED/30/790/2012 | Vatsala Thirumalai |
| Science and Engineering Research Board | EMR/2015/000595 | Vatsala Thirumalai |
| Department of Atomic Energy, Government of India | 12-R&DTFR-5.04-0800 | Vatsala Thirumalai |
| CSIR-UGC-NET | UGC Fellowship | Shaista Jabeen |
| Science and Engineering Research Board | YSS/2015/000908 | Gnaneshwar Yadav |

The funders had no role in study design, data collection and interpretation, or the decision to submit the work for publication.

### Author contributions

Sahana Sitaraman, Data curation, Formal analysis, Investigation, Visualization, Methodology, Writing - original draft, Writing - review and editing; Gnaneshwar Yadav, Data curation, Formal analysis, Funding acquisition, Investigation, Visualization, Methodology, Writing - original draft, Writing - review and editing; Vandana Agarwal, Shivangi Verma, Formal analysis, Investigation; Shaista Jabeen, Meha Jadhav, Investigation; Vatsala Thirumalai, Conceptualization, Data curation, Formal analysis, Supervision, Funding acquisition, Writing - original draft, Project administration, Writing - review and editing

### Author ORCIDs

Sahana Sitaraman [iD] https://orcid.org/0000-0003-4093-0619
Gnaneshwar Yadav [iD] https://orcid.org/0000-0002-1723-2831
Shaista Jabeen [iD] https://orcid.org/0000-0002-1888-2836
Shivangi Verma [iD] https://orcid.org/0000-0002-5245-9425
Meha Jadhav [iD] https://orcid.org/0000-0002-8898-1434
Vatsala Thirumalai [iD] https://orcid.org/0000-0002-2318-5023

### Ethics

Animal experimentation: Institutional Animal Ethics and Biosafety committee approvals were obtained for all procedures adopted in this study (NCB/IAEC/VT-1/2011 and TFR/NCBS/14-IBSC/VT-1/2011). Larvae and adults were reared using standard procedures (Westerfield, 2000).

### Decision letter and Author response

Decision letter https://doi.org/10.7554/eLife.68124.sa1
Author response https://doi.org/10.7554/eLife.68124.sa2

## Additional files

### Supplementary files

• Supplementary file 1. Fish-level synapse density (mean ± SEM) measurements in wild type and mutants.

• Supplementary file 2. Single-cell electroporation into Purkinje neurons (PNs) or non-PNs to identify electrically coupled neurons. When PNs were electroporated (animal nos. 1–8), no other dye-coupled neurons were detected. When non-PNs were electroporated (animal nos. 9–13), several other dye-coupled cells were seen, of which a couple were PNs. See also *Figure 7—figure supplement 1*.

• Supplementary file 3. A summary of the statistical analysis for data presented in the respective figures.

• Supplementary file 4. Summary of experimental manipulations, advantages, caveats, controls for caveats, and conclusions.

• Transparent reporting form

### Data availability

All data generated or analysed during this study are included in the manuscript and supporting files.

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
