## [Decision Letter]

**Acceptance summary:**

Gap junctions form between neurons throughout the brain and can provide electrical coupling between neurons, thereby coordinating their activity. Coordinated activity in pre- and postsynaptic neurons is thought to enhance synapse formation. This study demonstrates a role for gap junctions in the development of chemical glutamatergic synapses and dendritic arbor development in Zebrafish cerebellar Purkinje neurons, providing mechanistic insight into early stages of synaptogenesis.

**Decision letter after peer review:**

Thank you for submitting your article "Gjd2b-mediated gap junctions promote glutamatergic synapse formation and dendritic elaboration in Purkinje neurons" for consideration by *eLife*. Your article has been reviewed by 3 peer reviewers, one of whom is a member of our Board of Reviewing Editors, and the evaluation has been overseen by Didier Stainier as the Senior Editor. The reviewers have opted to remain anonymous.

The reviewers have discussed the reviews with one another and the Reviewing Editor has drafted this decision to help you prepare a revised submission.

Summary:

Sitaraman et al., reveal in their manuscript that cerebellar Purkinje neurons (PN) contain gap junctions based on Gjd2b expression prior to and accompanying chemical synapse formation. The authors use morpholino (MO) knockdowns and gjd2b-mutant larval zebrafish to explore the contributions that Gjd2b (and therefore neural gap junctions) makes during dendrite elaboration and synapse formation in Purkinje neurons (PNs). As consequence of these loss of function approaches the number of gap junctions in the cerebellum is reduced which is claimed to lead to a decrease of glutamatergic synapses in PNs. This is approached by electrophysiological studies and further suggested by ultrastructural analysis. Furthermore, a reduced dendritic branch outgrowth is found in gjd2b-mutant PNs, suggesting that synapses are placed closer to the soma in these mutant PNs. Dendrite outgrowth defects can be rescued by functional but not by non-functional Gjd2b-containing gap junctions in PNs, supporting a role for Gjd2b in PN dendrite outgrowth regulation

Furthermore, a correlation between CaMKII signaling, dendritic expansion, and Gjd2b is reported, and pharmacological inhibition of CaMKII can restore the dendritic expansion of the PCs.

The three reviewers agree that the manuscript's concept is novel and essential for the understanding of the cerebellar networks and particular of Purkinje cell development.

For example, the generation of the knocked-down/-out zebrafish where the gap junctions are eliminated is an excellent tool for the cerebellar development and network organization's dissection. Yet, all three reviewers also expressed that the manuscript is too premature to warrant publishing in its current form and that substantial additional data are required.

Essential revisions:

Morphants: The MO experiments are carefully executed and properly controlled, with a clear confirmation of protein knockdown. But the subsequent electrophysiological analysis requires further characterization. For example, a conclusive negative result in the PPR experiment (Figure 2D) is not clearly seen. A consistent separation between the groups across three ISIs from 100-200ms is shown, with an adjusted p value of 0.08 with an n of only 5. This looks like a preliminarily positive result, not a conclusively negative one. The experimental n should be increased to better test the null hypothesis, and/or the alternative interpretation should be presented and discussed.

Also, the interpretation of the peak amplitude needs attention, as the cumulative distributions show different shapes. A greater proportion of control events reach 15pA, but a larger proportion of morphant events reach 40pA. Clear evidence for increased amplitude from these data cannot be deduced.

Given that data from mutant analysis is stronger, the authors could consider to move the morphant analysis to the supplementary part. If the morphant analysis remains in the main body of the manuscript, data in Figure 1 and Figure 3 should be presented with the same scale values to allow for a better comparison between wild type, knock-down and knock-out data.

TALEN mutants: Immunohistochemistry against Cx35/36 clearly shows expression in the membrane of PNs (Figure S1B). In homozygous gjd2b-mutants immunoreactivity against Cx35/36 is not absent but reduced in the cerebellum (Figure S3E, to show exemplary images in addition would be helpful). This is explained by the antibody recognizing both Gjd2a and Gjd2b. It remains unclear then, whether Gjd2b is expressed in PNs. Likewise, if Cx35/36 staining remains in homozygous gjd2b-mutants due to parallel Gjd2a expression, then PNs should still contain a number of gap junctions. This needs to be clarified (see also dye-coupling experiments).

A nice approach is the rescue of dendrite outgrowth by PN-specific overexpression of Gjdb-mCherry in gjdb mutants. This is the only cell-autonomous approach presented. Does this approach also rescue the number of glutamatergic synapses? How about a rescue of electrophysiological properties of PNs?

Finally, it would be interesting to know whether any gross functional consequences/behavioral abnormalities have been observed in homozygous gjdb-mutants.

Electrophysiological recordings: It remains unclear how differences in voltage clamp recordings can be assigned to a reduced number of glutamatergic synapses. Inhibitors should be used to support that the recording inputs are indeed glutamatergic.

An alternative explanation is that the number of synapses could remain the same, but the release is affected; thus, gap junctions can modulate the synaptic strength.

The reported absence of NMDA mediated responses should be supported by data and control experiments. Can the authors record from a PC and apply NMDA in TTX? If there is no increase in the activity, this will be a piece of strong evidence.

Regarding climbing fiber stimulation, the authors should provide further evidence for the specificity of their stimulation to exclude that parallel fibers are not activated as well.

EM analysis: This is a labor-intensive approach that was nicely performed. Yet, the ultrastructural analysis of synapse density and the synapse maturation index in the molecular layer of the cerebellum is not PN specific. A potential contribution of eurydendroid cells to the reduction of synapse number and mature synapses should be taken into account. Also, it remains unclear whether the reduction in synapses is specific to glutamatergic synapses. An immunohistochemical analysis of excitatory synapses e.g. by PSD95-staining in wildtype versus gjd2b-mutants could reveal both whether the number of excitatory synapses in PNs is reduced and whether remaining synapses are indeed placed closer to the soma. Both images for controls and mutants should be shown.

Understanding the labor intensity of EM, and the 1.2um spacing that avoids resampling the same synapses, these data cannot be analysed as an n of (for example) 637 slices. This is an n of 3 animals. Strictly speaking, this is the only analysis that should be done (average value for each animal as a data point). Minimally, these fish-level data should be shown (as a scatter plot) alongside the cumulative probabilities across the numerous slices, with accompanying statistics.

Dye-coupling experiments: Electroporation of rhodamine dextran and neurobiotin into PNs does not cross to other neurons, although electroporation into non PNs occasionally labels PNs. This is not an especially strong argument for functional gap junctions. In addition, dye coupling in knocked-down and knocked-out animals should be shown to verify that the animals are indeed missing gap junctions in Purkinje cells. In addition, it remains unclear whether gap junctions are unidirectional and which cell type is electrically coupled to PNs.

CaMKII-signaling: The expression analysis regarding the relationship between gap junctions and CaMKII signalling is intriguing as it suggests that loss of gap junctions increases CaMKII signalling, which is in turn responsible for stabilizing the dendrites in a smaller/stunted form. This analysis though was performed on total RNA extractions of the entire brain, which does not resolve expression in Purkinje neurons, in addition CaMKII activity could also be regulated at the translational and posttranslational level. While the successful KN-93 rescue places CaMKII signaling downstream of Gjd2b, it could occur in a cell-autonomous or non-cell-autonomous fashion. These investigations lack information about direct or indirect functions of CaMKII and should be considered to be left out.

Statistics: All statistical analysis values are missing, e.g., U value, F values and degrees of freedom, and the actual and adjusted P values. The asterisks are just representative of the actual values and cannot replace the actual statistics.

[Editors’ note: the authors submitted for reconsideration following the decision after peer review. What follows is the decision letter after the second round of review.]

Thank you for resubmitting your work entitled "Gjd2b-mediated gap junctions promote glutamatergic synapse formation and dendritic elaboration in Purkinje neurons" for consideration by *eLife*. Your revised article has been reviewed by 3 peer reviewers, one of whom is a member of our Board of Reviewing Editors, and the evaluation has been overseen by a Senior Editor. The reviewers have opted to remain anonymous.

We are sorry to say that, after consultation with the reviewers, we have decided that your work will not be considered further for publication by *eLife*.

In order to provide orientation about the points of concern and strategies for adding experimental insight, I am listing below a collection of specific concerns raised by individual reviewers:

Immunohistochemistry and TEM data:

The authors carefully state that Gjd2b levels are reduced in morphants and mutants (e.g. line 131 or line 158), but this reduction (and not the complete absence) is difficult to understand if it is not mentioned initially that the anti-mouse Cx35/36 antibody cross reacts with Gjd2a and Gjd2b. This should be made clear from the beginning. Accordingly, line 116 should state "We confirmed that Gjd2a/b puncta..".

The authors report that anti-PSD95 immunohistochemistry analysis results in a higher number of synapse-counts on PNs that is contradictory to their results from TEM analysis and explain this with a limit in resolution by light microscopy (line 231) that does not allow one to unequivocally assign synapses to PNs. But should this erroneously assigning of synapses to PNs be the same in wild type and mutants? Why should this effect occur more often in the gjd2b mutants? This argument remains unclear.

The new PSD95 results are concerning. They are among the strongest results in the paper, and while the authors give two caveats associated with interpreting these data, the overall picture is not, on the whole, supportive of their interpretations.

I do not see that fish-level data have been added to Figure 5, where box plots continue to show data from hundreds of micrographs drawn from a small number of fish.

Electrophysiology:

I remain unconvinced that a conclusive negative result is shown for PPR in Figure 4 (D-G). There is a nonsignificant drop in PPR in the morphants and a sometimes-significant rise in PPR at short ISIs in the mutants. I understand that the distributions overlap, and that this is why most results are not significant, but I do not believe that the results and the experimental n are sufficient to support the claim that this is a negative result. I do not believe that this argument has changed appreciably since the initial submission

The authors suggest that the increase in peak amplitude and faster kinetics of mEPSCs in gjd2b mutants results from excitatory synapses being placed closer to the PN soma, implying a stunted dendritic arbor and loss of distal synapses. This line of arguments is difficult for me to understand. Why should the loss of distal excitatory synapses lead to increases in peak amplitude?

Or do the authors suggest that the stunted dendrites represent "compressed" dendrites in which synapses are moved closer to the soma? Then synapse density close to the soma should be investigated rather than distal synapses.

Then the analysis reveals a reduced total dendritic branch length in gjd2b mutant PNs and confirms a stunted dendritic arbor from which the authors conclude that fewer synapses are placed proximally to the PN somata in gjd2b mutants (line 260). Is this not contradictory to the argument presented in line 240? For me this was confusing and I suggest to reword this paragraph. Did the immunohistochemistry analysis against PSD95 reveal an insight into the localization of excitatory synapses closer to PN somata as suggested in the previous review?

Dye-coupling experiments:

These seem inconclusive, and I am not convinced that the reviewer's concerns have been addressed.

Presentation of Data

The authors carefully state that Gjd2b levels are reduced in morphants and mutants (e.g. line 131 or line 158), but this reduction (and not the complete absence) is difficult to understand if it is not mentioned initially that the anti-mouse Cx35/36 antibody cross reacts with Gjd2a and Gjd2b. This should be made clear from the beginning. Accordingly, line 116 should state "We confirmed that Gjd2a/b puncta..".

The authors state: "The scales for plots in Figures 1 and 2 have been matched.". I actually question that. First, I think that the Authors refer to Figures 1 and 3 as requested. The scale bar for Figure 3A, I doubt that is 10ms. I think it is 10 sec. Accordingly, in Figure 1A is 5pA, while in Figure 3A is 10pA. Finally, why is it so difficult to add the scale numbers in Figure3 as they have them in Figure 1 to compare the two figures directly? From this, I think the authors should be more careful regarding their statements and care more about their work presentation.

In the discussion (line 365-370) the authors emphasize their findings that glutamatergic synapses are decreased based on structural and physiological data but do not mention their contradictory finding with PSD95 immunohistochemistry. This should be discussed more cautiously.

"The authors present the holding potential as the membrane potential, which is confusing to distinguish between EPSPs and EPSCs." I referred to figure 4, where the authors in Panel B write the mV at the beginning of the traces that is a common practice for current-clamp recordings. In voltage-clamp recordings, there is the 0pA. The holding is usually added above, below, or in legend and refer as that (e.g., Holding at -65 mV). To this end, I also noticed that for Figures 1 and 3, any reference to holding potential is missing. Is that again the -65mV or different?

The authors should try to stimulate the inferior olive to specifically address the input of climbing fibers.

The authors should discuss the discrepancy between their findings and the observation of gap junction coupling between cerebellar neurons recently found in adult zebrafish (Chang et al., 2020). That is important as in most neuronal networks; the gap junctions appear earlier, and they are more abundant in earlier developmental stages than the chemical synapses.

Summary:

Thanks a lot for having submitted a revised version of your manuscript entitled "Gjd2b-mediated gap junctions promote glutamatergic synapse formation and dendritic elaboration in Purkinje neurons". This revised version has now been seen by three reviewers and they share the opinion that the manuscript's concept is novel and essential for understanding cerebellar circuitry development. Also, all three reviewers were pleased by the new experimental data and changes to the manuscript that have been added by the authors. The manuscript has been improved addressing some of the concerns that have been raised during the initial review. Yet, the different reviews also coincide in the view that the presented data still fall short in being consistently convincing and that the presented study remains premature at this point and regretfully lacks sufficient support for publishing this experimental study.

*Reviewer #1:*

Reading the revised version of the manuscript "Gjd2b-mediated gap junctions promote glutamatergic synapse formation and dendritic elaboration in Purkinje neurons" by Sitaraman et al., I am pleased by the new experimental data and changes to the manuscript that have been added by the authors. The data is in most parts presented clearly, but a few points should be addressed to make some of their arguments more clear or to avoid misunderstandings of some of their claims.

1) According to the guidelines of *eLife* the title of the manuscript should mention the use of zebrafish as model system e. g.: "Gjd2b-mediated gap junctions promote glutamatergic synapse formation and dendritic elaboration in zebrafish Purkinje neurons"

2) The authors carefully state that Gjd2b levels are reduced in morphants and mutants (e.g. line 131 or line 158), but this reduction (and not the complete absence) is difficult to understand if it is not mentioned initially that the anti-mouse Cx35/36 antibody cross reacts with Gjd2a and Gjd2b. This should be made clear from the beginning. Accordingly, line 116 should state "We confirmed that Gjd2a/b puncta..".

3) Line 221: the reduced number of synapses observed in TEM recordings could suggest but does not indicate a reduced number of formed synapses on PNs as this analysis is not able to distinguish between PNs and e.g. eurydendroid cells. I suggest to reword this sentence into "..these results indicate that Gjd2b is involved in the formation of synapses in the molecular layer of the cerebellum to which PN synapses significantly contribute.."

4) The authors report that anti-PSD95 immunohistochemistry analysis results in a higher number of synapse-counts on PNs that is contradictory to their results from TEM analysis and explain this with a limit in resolution by light microscopy (line 231) that does not allow one to unequivocally assign synapses to PNs. But should this erroneously assigning of synapses to PNs be the same in wild type and mutants? Why should this effect occur more often in the gjd2b mutants? This argument remains unclear to me.

5) Line 240: the authors suggest that the increase in peak amplitude and faster kinetics of mEPSCs in gjd2b mutants results from excitatory synapses being placed closer to the PN soma, implying a stunted dendritic arbor and loss of distal synapses. This line of arguments is difficult for me to understand. Why should the loss of distal excitatory synapses lead to increases in peak amplitude?

Or do the authors suggest that the stunted dendrites represent "compressed" dendrites in which synapses are moved closer to the soma? Then synapse density close to the soma should be investigated rather than distal synapses.

Then the analysis reveals a reduced total dendritic branch length in gjd2b mutant PNs and confirms a stunted dendritic arbor from which the authors conclude that fewer synapses are placed proximally to the PN somata in gjd2b mutants (line 260). Is this not contradictory to the argument presented in line 240? For me this was confusing and I suggest to reword this paragraph. Did the immunohistochemistry analysis against PSD95 reveal an insight into the localization of excitatory synapses closer to PN somata as suggested in the previous review?

6) In the discussion (line 365-370) the authors emphasize their findings that glutamatergic synapses are decreased based on structural and physiological data but do not mention their contradictory finding with PSD95 immunohistochemistry. This should be discussed more cautiously.

7) Also, the authors should point out in the discussion that currently besides the cell type specific rescue of PN dendrite outgrowth in gjd2b mutants, they can currently not distinguish between cell-autonomous and non-cell autonomous effects of gjd2b loss on PNs. This should be made clear to the reader.

*Reviewer #2:*

The manuscript is now improved in several aspects. The authors performed additional experiments to verify their observations. They followed most of the reviewers' recommendations, yet the overall impression is that the Authors did not consider some critical comments. Specifically, the ones that aimed to improve the data's presentation.

Specifically:

The authors state: "The scales for plots in Figures 1 and 2 have been matched.". I actually question that. First, I think that the Authors refer to Figures 1 and 3 as requested. The scale bar for Figure 3A, I doubt that is 10ms. I think it is 10 sec. Accordingly, in Figure 1A is 5pA, while in Figure 3A is 10pA. Finally, why is it so difficult to add the scale numbers in Figure3 as they have them in Figure 1 to compare the two figures directly? From this, I think the authors should be more careful regarding their statements and care more about their work presentation.

Regarding the question of the specificity of the CF stimulation, the answer is not convincing. The fact that the authors used this approach in the past is not enough to claim specific. Why did the authors not try to stimulate the inferior olive?

Regarding the Dye-coupling experiments where the authors state that they do not observe any dye coupling between PCs. In light of the new paper published in PNAS (Chang et al., 2020), the authors should discuss this discrepancy between their findings and what is observed in adult zebrafish. That is important as in most neuronal networks; the gap junctions appear earlier, and they are more abundant in earlier developmental stages than the chemical synapses.

Regarding our previous comment: "The authors present the holding potential as the membrane potential, which is confusing to distinguish between EPSPs and EPSCs." I referred to figure 4, where the authors in Panel B write the mV at the beginning of the traces that is a common practice for current-clamp recordings. In voltage-clamp recordings, there is the 0pA. The holding is usually added above, below, or in legend and refer as that (e.g., Holding at -65 mV). To this end, I also noticed that for Figures 1 and 3, any reference to holding potential is missing. Is that again the -65mV or different?

Recommendations for the authors:

The manuscript is now improved in some aspects; however, the overall impression that I have is that the Authors did not take into consideration all the comments, and I found somehow quite lousy the revision of the previous manuscript. Specifically:

The authors state: "The scales for plots in Figures 1 and 2 have been matched.". I actually question that. First, I think that the Authors refer to Figures 1 and 3 as requested. The scale bar for Figure 3A, I doubt that is 10ms. I think it is 10 sec. Accordingly, in Figure 1A is 5pA, while in Figure 3A is 10pA. Finally, why is it so difficult to add the scale numbers in Figure3 as they have them in Figure 1 to compare the two figures directly? From this, I think the authors should be more careful regarding their statements and care more about their work presentation.

Regarding the question of the specificity of the CF stimulation, the answer is not convincing. The fact that the authors used this approach in the past is not enough to claim specific. Why did the authors not try to stimulate the inferior olive?

Regarding the Dye-coupling experiments where the authors state that they do not observe any dye coupling between PCs. In light of the new paper published in PNAS (Chang et al., 2020), the authors should discuss this discrepancy between their findings and what is observed in adult zebrafish. That is important as in most neuronal networks; the gap junctions appear earlier, and they are more abundant in earlier developmental stages than the chemical synapses.

Regarding our previous comment: "The authors present the holding potential as the membrane potential, which is confusing to distinguish between EPSPs and EPSCs." I referred to figure 4, where the authors in Panel B write the mV at the beginning of the traces that is a common practice for current-clamp recordings. In voltage-clamp recordings, there is the 0pA. The holding is usually added above, below, or in legend and refer as that (e.g., Holding at -65 mV). To this end, I also noticed that for Figures 1 and 3, any reference to holding potential is missing. Is that again the -65mV or different?

*Reviewer #3:*

As outlined in my original review of this manuscript, I view it as interesting and potentially impactful, but preliminary in its interpretations and not consistently convincing. I do not view this has having changed in a meaningful way with these revisions. The new data provided do not particularly strengthen conclusions, and in one case conflict with the manuscript's narrative. Most reviewers' comments have been addressed without further experiments, but not in a way that is consistently satisfying. I address these revisions below, broken down by sections of the rebuttal letter.

Morphants:

I remain unconvinced that a conclusive negative result is shown for PPR in Figure 4. There is a nonsignificant drop in PPR in the morphants and a sometimes-significant rise in PPR at short ISIs in the mutants. I understand that the distributions overlap, and that this is why most results are not significant, but I do not believe that the results and the experimental n are sufficient to support the claim that this is a negative result. I do not believe that this argument has changed appreciably since the initial submission.

I accept the authors' assertion that increased peak amplitude for the mutant is convincingly demonstrated in Figure 3D.

I continue to think that the mutant provides stronger support for the authors' claims than the MOs do.

TALEN Mutants:

I am generally convinced by the authors' responses to these questions and with the added data, although I defer to the reviewer who originally raised the issue of the two orthologs with regard to whether this has been adequately addressed.

Electrophysiological recordings:

As described above, I continue not to be convinced of a negative result for the PPD data shown in Figure 4D-G.

The arguments presented about blockers and the new NMDA pulse data appear to be valid to me, but I defer to the relevant reviewer.

EM Analysis:

I understand the converging lines of evidence that the authors refer to (ephys and EM), and understand that these could be viewed as complementary, given each line's strengths and caveats. This does not really address the reviewer's concern, although I defer to him/her with regard to whether they are convinced.

The new PSD95 results are concerning. They are among the strongest results in the paper, and while the authors give two caveats associated with interpreting these data, the overall picture is not, on the whole, supportive of their interpretations.

I do not see that fish-level data have been added to Figure 5, where box plots continue to show data from hundreds of micrographs drawn from a small number of fish.

Dye-coupling experiments:

These seem inconclusive, and I am not convinced that the reviewer's concerns have been addressed.

CaMKII-signalling:

This was an interesting but not fully supported element of the original manuscript. I agree with the decision to withdraw these data, but it leaves a less impactful paper.

[Editors' note: further revisions were suggested prior to acceptance, as described below.]

Thank you for submitting your article "Gjd2b-mediated gap junctions promote glutamatergic synapse formation and dendritic elaboration in Purkinje neurons" for consideration by *eLife*. Your article has been reviewed by 3 peer reviewers, including Hollis T Cline as the Reviewing Editor and Reviewer #1, and the evaluation has been overseen by Didier Stainier as the Senior Editor. The following individual involved in review of your submission have agreed to reveal their identity: Alanna J Watt (Reviewer #3).

Essential Revisions:

Please modify the text of the paper to identify caveat, alternate interpretations and open questions for future research, as suggested in the reviewers comments, below.

*Reviewer #1 (Recommendations for the authors):*

1. It would be more clear to name the splice-blocking morpholino as 'Gjd2b-MO'

2. Control morpholinos were tested separately in experiments using immunolabeling, but were not used in the electrophysiology experiments. This is unusual but acceptable.

3. Figure 1. Supplement 1. Please label columns and rows in panel B.

4. For morphants, please state if experimental and control animals were from the same clutch and how many clutches of embryos were used for each experiment.

5. Figure 4. Panel B, list specific ages in legend, rather than 'mixed ages'. Panel C, state that this is done on the presence of TTX.

6. In the first paragraph of discussion, authors should more clearly state that they think that conductance through Cx36/Gjd2b-mediated gap junctions is not be sufficient to pass dye and they are therefore not able to identify cells coupled to PCs.

7. Given the ambiguity about the sequence of events governing branch extension gap junction formation -> synaptogenesis->branch extension, repeat versus gap junction formation -> branch extension-> synaptogenesis, repeat., the interpretation on line 242 'resulting in fewer synapses' is overstated.

8. The authors state that it is likely that gap junctions promote dendritic arbor growth directly, independent of their actions on chemical synapses. What evidence (citation) supports this statement?

9. On pg 19 the first full paragraph is repeated in the next section.

10. Figure 7, supplement 1. There is a problem with the labeling

*Reviewer #2 (Recommendations for the authors):*

Electrophysiology data:

1. The data on mEPSC frequency is clear evidence for fewer excitatory synapses. However, interpreting larger amplitudes and faster decays as an explanation for the loss of distal synapses is problematic. The shift towards larger amplitudes can also be explained simply by the absence of weaker synapses rather than arguing for the absence of distal synapses. Moreover, larger amplitudes can also be a homeostatic scaling mechanism to compensate for fewer synapses. Regarding the decay, this evidence would be more compelling if these experiments were done in current clamp where the absence of gap junctions would lead to longer voltage decay as a result of increased input resistance of neurons. In contrast, the faster current decay observed by the authors could also be a result of different subunit composition or other biophysical considerations (Laurence et al., Nature Neuroscience, 2005; Kumar et al., JNeuroscience 2002), and unless that is ruled out explicitly, I would recommend adding this as an important alternative explanation in the discussion.

2. What is the input resistance of mutant neurons versus wildtype Purkinje neurons? Given the smaller dendritic arbors and the absence of gap junctions, one would expect Gjdb2-/- neurons have a higher input resistance. It will be good to see these data.

3. In comparing the WT data in Figure 1D,E,F with Figure 3D,E,F, I am confused by differences in the distribution of WT data between these figures. There is a long tail in the distribution of WT data in Figure 1D,E,F which is absent in Figure 3D,E,F. I am curious why this is the case.

EM data:

1. I agree with the other reviewers that while the EM data are beautiful and hard to collect and analyze, not being able to attribute synapses to PNs significantly limits the conclusions one can reach from this experiment. I do not know much about cerebellar circuitry but is there any estimate for what fraction of excitatory synapses are formed on PNs versus other neurons? Also, the inability to distinguish between excitatory and inhibitory synapses is a major limitation. Between the EM and electrophysiology data, I would argue that the electrophysiology data are far stronger. I understand that the authors see this as converging evidence, but in my opinion, the EM data have substantial caveats and at best provide weak support for the conclusions the authors are trying to reach. If, for instance, the authors have data on spontaneous IPSCs in mutant and WT neurons that are similar in frequency, that can at least help argue that the numbers of inhibitory synapses are similar.

2. I have not seen the PSD95 data, but I agree with the authors that PSD95 staining is not a compelling experiment. Light microscopy resolution is a major challenge. If anything, the authors could have tried expressing PSD95-tagged GFP in individual neurons in mutant and WT fish as was done in Niell et al. (Nature Neuroscience 2004) in the zebrafish optic tectum. I am not suggesting that the authors do this experiment but wanted to just throw in support for their argument that PSD95 staining is inconclusive.

Dendritic elaboration and CaMKII

1. The gain of function experiments are interesting, but perhaps I am missing something here. My understanding is that functional electrical synapses need the assembly of a pore in the presynaptic and post-synaptic neuron. So, how would expressing Gjd2b in one neuron ensure functional electrical connectivity with other neurons? I see this is addressed in the limitations document, but the authors' reliance on pore dead experiments is unconvincing. If anything, the pore dead neurons have dendrites that are comparable to WT neurons and longer than the Gjd2b neurons (Figure 7b). I think the authors need to do a statistical analysis of differences between WT and Gjd2b rescue as well as between WT and Gjd2b pore-dead mutants. I think there is something interesting there that might allude to functional aspects of non-pore forming regions of Gjd2b. In the absence of clear experiments to demonstrate functional electrical synapses, I think this experiment falls significantly short of implicating electrical synapses in dendrite elaboration. If the authors do want to make this claim, in the very least they need to show that the "rescued" neurons have comparable excitatory synapses

2. A lot of the work on dendritic elaboration has parallels with the rich body of work done in Hollis Cline's lab which the authors reference extensively. However, it's not clear to me that the dendritic effects are not simply a downstream consequence of the absence of Gjd2b rather any information transmitted through the electrical synapses. Since Gjd2b knockout reduces the number of AMPAR synapses based on the electrophysiology, isn't a simple explanation for all the dendritic effects simply a consequence of fewer AMPAR synapses as shown in Haas et al. (PNAS, 2006). Moreover, given the lack of direct evidence that the rescue experiments lead to functional electrical synapses, I am not convinced that molecules transmitted through gap junctions are somehow responsible for elevated CaMKII.

In summary, while this paper represents a substantial amount of work and relies on converging lines of evidence to arrive at their conclusions, there are several limitations within each technique and these shortcomings are not addressed by the complimentary experiments.

The authors present good evidence for fewer chemical synapses and shorter dendrites in Purkinje neurons in fish where Gjd2b dependent electrical synapses are knocked down or knocked out. The concerns about electrophysiology data can be addressed in the discussion as an important caveat.

However, my bigger concern is with disambiguating Gjd2b mediated changes in dendritic structure from downstream effects of simply having fewer AMPAR synapses. Previous work has provided compelling evidence that chemical transmission through AMPAR synapses is a key driver of dendritic elaboration. So, if there are fewer AMPAR synapses, is it not unsurprising that the dendrites are smaller and that has nothing to do directly with Gjd2b function? Perhaps I am missing a key piece of the argument here and would be happy to be proven wrong.

*Reviewer #3 (Recommendations for the authors):*

Having read through the previous response to the reviewers, I think that the authors have addressed them very well. I do not think further experiments or analyses are required.

---

## [Author Response]

Essential revisions:Morphants: The MO experiments are carefully executed and properly controlled, with a clear confirmation of protein knockdown. But the subsequent electrophysiological analysis requires further characterization. For example, a conclusive negative result in the PPR experiment (Figure 2D) is not clearly seen. A consistent separation between the groups across three ISIs from 100-200ms is shown, with an adjusted p value of 0.08 with an n of only 5. This looks like a preliminarily positive result, not a conclusively negative one. The experimental n should be increased to better test the null hypothesis, and/or the alternative interpretation should be presented and discussed.Also, the interpretation of the peak amplitude needs attention, as the cumulative distributions show different shapes. A greater proportion of control events reach 15pA, but a larger proportion of morphant events reach 40pA. Clear evidence for increased amplitude from these data cannot be deduced.Given that data from mutant analysis is stronger, the authors could consider to move the morphant analysis to the supplementary part. If the morphant analysis remains in the main body of the manuscript, data in Figure 1 and Figure 3 should be presented with the same scale values to allow for a better comparison between wild type, knock-down and knock-out data.

To conclusively say if the paired pulse ratios in PNs differ after Gjd2b manipulation, we decided to repeat the PPD experiment in WT and gjd2b-/- zebrafish larvae. We recorded climbing fibre evoked responses from PNs in 7dpf larvae, at inter stimulus intervals varying from 30-1000 ms (for details, see Materials and methods). We observe robust paired pulse depression in wildtype PNs, as is seen by the significant difference in paired pulse ratios at 30,35,40,50,70 ms inter-stimulus intervals in comparison to all longer intervals (Figure 4). We also do not observe any significant difference between wildtype and mutant PPRs at any of the ISIs, except at 40ms (p value 0.03). Though average values of the two genotypes show separation, the distributions are largely overlapping. This indicates that Gjd2b manipulation either by knock-out or knock-down approaches does not affect the neurotransmitter release probability at climbing fibre-Purkinje neuron synapses.

The peak amplitude of mEPSCs shows an increase in morphants compared to control, as seen by a shift to the right of the morphant distribution relative to control for about 90% of all events. This can also be seen in the box plots that we have now added to this figure. We disagree that this is not clear evidence for an increase in amplitude.

The scales for plots in Figure 1 and 2 have been matched.

TALEN mutants: Immunohistochemistry against Cx35/36 clearly shows expression in the membrane of PNs (Figure S1B). In homozygous gjd2b-mutants immunoreactivity against Cx35/36 is not absent but reduced in the cerebellum (Figure S3E, to show exemplary images in addition would be helpful). This is explained by the antibody recognizing both Gjd2a and Gjd2b. It remains unclear then, whether Gjd2b is expressed in PNs. Likewise, if Cx35/36 staining remains in homozygous gjd2b-mutants due to parallel Gjd2a expression, then PNs should still contain a number of gap junctions. This needs to be clarified (see also dye-coupling experiments).A nice approach is the rescue of dendrite outgrowth by PN-specific overexpression of Gjdb-mCherry in gjdb mutants. This is the only cell-autonomous approach presented. Does this approach also rescue the number of glutamatergic synapses? How about a rescue of electrophysiological properties of PNs?Finally, it would be interesting to know whether any gross functional consequences/behavioral abnormalities have been observed in homozygous gjdb-mutants.

Corroborating our IHC results, the study by Takeuchi et al., 2016 shows that both Gjd2a and Gjd2b transcripts are expressed in zebrafish Purkinje neurons (https://doi.org/10.1002/cne.24114). This study looks at the transcript levels of various genes in 4 cell types of the zebrafish cerebellum: granule cells, Purkinje neurons, eurydendroid cells and Bergmann glia; and also in neurons of the inferior olive. All 3 biological replicates of Purkinje neuron samples show expression of the Cx35b/Gjd2b transcript.

Exemplar images on Cx35 IHC from WT and mutant larvae have been included in figure 2.

We acknowledge that knocking out Gjd2b will not rid PNs of all their gap junctions. Keeping that in mind, we only claim that Gjd2b gap junctions affect glutamatergic synapse formation and dendritic arborisation in PNs. There may be similar roles for other gap junction proteins not studied here.

While it would have been further confirmation of our results to demonstrate rescue of the number of glutamatergic synapses using TEM or electrophysiology, these are technically difficult and low yield experiments. The gjd2b-mCherry construct is expressed in a mosaic fashion in isolated PNs to ascertain cell-autonomous rescue and such neurons are difficult to target specifically with TEM or electrophysiology. In the manuscript, we make specific and limited claims regarding the rescue experiment with respect to the dendritic growth phenotype and do not extrapolate it to the number of glutamatergic synapses. We discuss a possible rescue approach using PSD95 IHC and its caveats in the section below.

Qualitatively, we do not observe any stark changes in the behaviour of gjd2b-/- fish, as compared to wildtype larvae. In-depth, quantitative assays to determine if finer behavioural aspects are affected by loss of Gjd2b is out of scope of this paper.

Electrophysiological recordings: It remains unclear how differences in voltage clamp recordings can be assigned to a reduced number of glutamatergic synapses. Inhibitors should be used to support that the recording inputs are indeed glutamatergic.An alternative explanation is that the number of synapses could remain the same, but the release is affected; thus, gap junctions can modulate the synaptic strength.The reported absence of NMDA mediated responses should be supported by data and control experiments. Can the authors record from a PC and apply NMDA in TTX? If there is no increase in the activity, this will be a piece of strong evidence.Regarding climbing fiber stimulation, the authors should provide further evidence for the specificity of their stimulation to exclude that parallel fibers are not activated as well.

All voltage clamp recordings were performed in the presence of a cocktail of blockers to block any GABA and NMDA mediated minis. This is mentioned in the ‘Electrophysiology’ section of methods, line 518-519, “mEPSCs were recorded after bath application of TTX (1μM), Gabazine (10μM) and APV (40μM)”. The recorded mEPSCs constitute AMPAR-mediated glutamatergic currents.

To check if a change in probability of release can explain the change in mEPSC frequency, we stimulated CFs and measured paired pulse ratios of the CF to PN synapse. We have demonstrated previously that stimulation of CFs evokes synaptic currents in PNs which have markedly higher amplitudes than those evoked by parallel fibers (*eLife* 2015;4:e09158). In both mutants (new data, figure 4F and G) and morphants, we failed to find significant changes in the paired pulse ratios when compared to control. These findings suggest that the observed increase in mEPSC frequency probably arises due to changes in synapse numbers, which we then confirm using ultrastructural analysis.

As suggested by the reviewers, we have now added new experiments where we pulsed NMDA and measured PN responses in wild type and mutant larvae. With TTX in the bath, brief puffs of NMDA applied to PNs held at -20mV failed to elicit any responses in wild type or mutant larvae (figure 4C).

EM analysis: This is a labor-intensive approach that was nicely performed. Yet, the ultrastructural analysis of synapse density and the synapse maturation index in the molecular layer of the cerebellum is not PN specific. A potential contribution of eurydendroid cells to the reduction of synapse number and mature synapses should be taken into account. Also, it remains unclear whether the reduction in synapses is specific to glutamatergic synapses. An immunohistochemical analysis of excitatory synapses e.g. by PSD95-staining in wildtype versus gjd2b-mutants could reveal both whether the number of excitatory synapses in PNs is reduced and whether remaining synapses are indeed placed closer to the soma. Both images for controls and mutants should be shown.Understanding the labor intensity of EM, and the 1.2um spacing that avoids resampling the same synapses, these data cannot be analysed as an n of (for example) 637 slices. This is an n of 3 animals. Strictly speaking, this is the only analysis that should be done (average value for each animal as a data point). Minimally, these fish-level data should be shown (as a scatter plot) alongside the cumulative probabilities across the numerous slices, with accompanying statistics.

We present complementary lines of evidence using electrophysiology and EM analysis for the change in synapse number following gjd2b manipulation. The electrophysiology experiment is specific for glutamatergic synaptic inputs and the decrease in mEPSC frequency is suggestive of decrease in synapse numbers. The EM analysis provides definitive counts of synapse numbers in the molecular layer where PNs elaborate dendrites and make synapses. Yet, the EM approach is not specific for PNs or for glutamatergic synapses. Taken together, these two complementary lines of evidence indicate a reduction in glutamatergic synapse density.

We performed immunohistochemical staining of PSD95 in wildtype and mutant larvae as suggested by the reviewers. We identified PNs with Parv7 or EGFP counterstaining. We quantified the numbers of PSD95 puncta colocalized with Parvalbumin-7/EGFP and compared the values between wild type and mutant larvae. We found that the number of PSD95 puncta was increased significantly in mutants compared to wild type (Figure 5 —figure supplement 1). While this result is inconsistent with our EM and electrophysiology results, it also suffers from these caveats:

1.Though we're only counting puncta which are colocalised with Parv7 positive neurons, the resolving limit of light microscopy can cause us to include puncta which are present in adjacent neurons as well. Therefore, this method does not guarantee specificity for synapses on PNs.

2. Presence of a PSD95 punctum does not guarantee the presence of a functional synapse. The immunostaining technique will label synaptic as well as non synaptic clusters of PSD95 that are either at nascent synapses or other intracellular membrane organelles (El-Husseini et al., J. Cell Biol., 2000). We do not know if loss of Gjd2b causes PSD95 to accumulate in these non-synaptic or immature synaptic clusters.

Including a pre-synaptic marker and performing super resolution microscopy will give more reliable results. But such a study is beyond the scope of this paper.

Seeing that this experiment did not recapitulate our EM results, we did not continue with assessing the rescue capabilities of Gjd2b-mCherry on glutamatergic synapses using PSD95 IHC.

Fish-level data for the TEM has been shown as a boxplot with scatter, alongside the cumulative probabilities, in Figure 5.

Dye-coupling experiments: Electroporation of rhodamine dextran and neurobiotin into PNs does not cross to other neurons, although electroporation into non PNs occasionally labels PNs. This is not an especially strong argument for functional gap junctions. In addition, dye coupling in knocked-down and knocked-out animals should be shown to verify that the animals are indeed missing gap junctions in Purkinje cells. In addition, it remains unclear whether gap junctions are unidirectional and which cell type is electrically coupled to PNs.

We used dye-coupling approaches to establish if PNs make functional gap junctions. We have tested this using neurobiotin (323 Da) and serotonin (176 Da). Both approaches gave us the same result, which is that when Purkinje neurons were filled with the dye, we did not observe any coupled neurons. In both experiments, electroporation of other cerebellar neurons resulted in coupled cells taking up the dye, demonstrating the effectiveness of our methodology. In all trials where non-PNs were electroporated, at least a couple of PNs showed uptake of the dye from the electroporated neuron (Figure 7 —figure supplement 1 and Supplementary file 1). While this is not confirmatory proof, it is suggestive of rectifying gap junctions and we present it as such in the manuscript. The identity of the cell type coupled to PNs awaits future experimentation.

The absence of coupling among PNs and the low numbers of coupled PNs observed after electroporating non-PNs preclude attempts to verify their absence or reduction in the mutants and morphants respectively.

CaMKII-signaling: The expression analysis regarding the relationship between gap junctions and CaMKII signalling is intriguing as it suggests that loss of gap junctions increases CaMKII signalling, which is in turn responsible for stabilizing the dendrites in a smaller/stunted form. This analysis though was performed on total RNA extractions of the entire brain, which does not resolve expression in Purkinje neurons, in addition CaMKII activity could also be regulated at the translational and posttranslational level. While the successful KN-93 rescue places CaMKII signaling downstream of Gjd2b, it could occur in a cell-autonomous or non-cell-autonomous fashion. These investigations lack information about direct or indirect functions of CaMKII and should be considered to be left out.

Acting on advice from the reviewers, we have decided to remove these data from this manuscript.

Statistics: All statistical analysis values are missing, e.g., U value, F values and degrees of freedom, and the actual and adjusted P values. The asterisks are just representative of the actual values and cannot replace the actual statistics.

All statistical values have been provided in the Results section. A separate statistical summary document is also provided.

[Editors’ note: the authors resubmitted a revised version of the paper for consideration. What follows is the authors’ response to the second round of review.]In order to provide orientation about the points of concern and strategies for adding experimental insight, I am listing below a collection of specific concerns raised by individual reviewers:Immunohistochemistry and TEM data:The authors carefully state that Gjd2b levels are reduced in morphants and mutants (e.g. line 131 or line 158), but this reduction (and not the complete absence) is difficult to understand if it is not mentioned initially that the anti-mouse Cx35/36 antibody cross reacts with Gjd2a and Gjd2b. This should be made clear from the beginning. Accordingly, line 116 should state "We confirmed that Gjd2a/b puncta..".

We mentioned this in the corresponding figure legend (line 971), and have made the suggested modification in the main text (lines 110, 121 and 125).

The authors report that anti-PSD95 immunohistochemistry analysis results in a higher number of synapse-counts on PNs that is contradictory to their results from TEM analysis and explain this with a limit in resolution by light microscopy (line 231) that does not allow one to unequivocally assign synapses to PNs. But should this erroneously assigning of synapses to PNs be the same in wild type and mutants? Why should this effect occur more often in the gjd2b mutants? This argument remains unclear.

The main issue with the EM data pointed out by the reviewers was that it was neither specific to PNs nor to glutamatergic synapses. However, the experiment suggested by the reviewers does not address either of those issues unequivocally:

The PSD95 puncta observed in the IHC experiment can be of three kinds:

1. Synaptic puncta on PNs

2. Non-synaptic puncta on PNs

3. Synaptic and non-synaptic puncta in non-PNs located very close to PNs.

Nevertheless, in an attempt to address reviewer concerns, we performed PSD95 immunohistochemistry with PN labeling and quantified PSD95 puncta on PNs in wild type and mutant larvae. We saw an increase in the number of PSD95 puncta in gjd2b mutants. While this may be taken to mean an increase in synaptic puncta in PNs in the mutants, the other equally likely possibilities are:

1. An increase in PSD95 accumulation at non-synaptic sites in PNs due to the mutation.

2. An increase in PSD95 accumulation in synaptic and/or non-synaptic sites in nonPNs also due to the mutation in Gjd2b.

Due to the limit in resolution of light microscopy, PSD95 puncta cannot be assigned to PNs with certainty. If the Gjd2b mutation affects PSD95 expression levels in neighbouring cells or promotes its accumulation intracellularly, this error in assignment will be greater in mutants compared to wild type.

Therefore, the results from the PSD95 experiment are inconclusive, while those from the electrophysiology and TEM analysis point clearly to a decrease in PN synapse numbers when Gjd2b is impaired. Since the new PSD95 data do not provide categorical answers and suffer from major caveats, we have removed this experiment from the current version.

The new PSD95 results are concerning. They are among the strongest results in the paper, and while the authors give two caveats associated with interpreting these data, the overall picture is not, on the whole, supportive of their interpretations.

We disagree that the PSD95 results are among the strongest in the paper. In our original submission, reviewers had two concerns in the EM data about the lack of specificity to either PNs or glutamatergic synapses. They suggested quantifying PSD95 puncta on PNs as a way of addressing these concerns. However, for reasons mentioned above, this experiment does not categorically answer either the specificity to PNs or that all puncta are glutamatergic synapses. In our earlier response, we clearly explained how the EM and the mEPSC data have to be taken together as complementary approaches, one ultrastructural and the other functional. We are extremely disappointed that the reviewers have discounted all of the other experiments presented in the manuscript and focused on one experiment with equal number of caveats. We do not believe that the PSD95 results have completely negated all of the other experimental results. We now include a table that shows the main result, interpretation and caveats of all the experiments presented in the manuscript (Supplementary file 4).

I do not see that fish-level data have been added to Figure 5, where box plots continue to show data from hundreds of micrographs drawn from a small number of fish.

As mentioned in the manuscript, the EM data were derived from sectioning the brains of 3 fish each in wild type and mutants. It will not be meaningful to plot 3 data points. We present fish level means and SEMs in the form of a table (Supplementary file 1).

Electrophysiology:I remain unconvinced that a conclusive negative result is shown for PPR in Figure 4 (D-G). There is a nonsignificant drop in PPR in the morphants and a sometimes-significant rise in PPR at short ISIs in the mutants. I understand that the distributions overlap, and that this is why most results are not significant, but I do not believe that the results and the experimental n are sufficient to support the claim that this is a negative result. I do not believe that this argument has changed appreciably since the initial submission

We respectfully disagree with the reviewer. The n’s we have for the PPR experiments are 14 for wild type PNs and 13 for the mutants. These n’s are typical for PPR experiments and perhaps even higher than what is reported in the literature. The p-value we obtained was 0.17, which is not even borderline. The distributions are overlapping and the statistical testing did not yield significance. We cannot reject the null hypothesis. We do not understand how a visual difference in the mean can override rigorous statistical analysis.

We want to point you to some recent papers in *eLife* where PPR experiments have been reported for comparisons on n’s used.

DOI: 10.7554/*eLife*.45920: Figure 2—figure supplement 2, n = 8 and 9 cells, ns, p = 0.08.

DOI: 10.7554/*eLife*.31755: Figure 4—figure supplement 4, n = 7 and 8 cells, ns, p value not reported.

DOI: 10.7554/*eLife*.36209: Figure 4: n = 5 and 6 cells, ns, p = 0.97

DOI: 10.7554/*eLife*.33892: Figure 2B and D: n = 6 cells, ns, p=0.156

The authors suggest that the increase in peak amplitude and faster kinetics of mEPSCs in gjd2b mutants results from excitatory synapses being placed closer to the PN soma, implying a stunted dendritic arbor and loss of distal synapses. This line of arguments is difficult for me to understand. Why should the loss of distal excitatory synapses lead to increases in peak amplitude?

mEPSCs arriving from synapses placed closer to the soma will be filtered less by the cable while those arriving from synapses on distal dendrites will be filtered more, resulting in slower kinetics and attenuated amplitudes. In neurons with stunted arbors, these distally placed synapses will be absent or reduced in number. Therefore, the distribution of mEPSCs moves towards larger amplitudes and faster kinetics.

Or do the authors suggest that the stunted dendrites represent "compressed" dendrites in which synapses are moved closer to the soma? Then synapse density close to the soma should be investigated rather than distal synapses.

No we are not suggesting this at all. There is no need to assume changes in synapse placement or synapse density on individual branches.

Then the analysis reveals a reduced total dendritic branch length in gjd2b mutant PNs and confirms a stunted dendritic arbor from which the authors conclude that fewer synapses are placed proximally to the PN somata in gjd2b mutants (line 260). Is this not contradictory to the argument presented in line 240? For me this was confusing and I suggest to reword this paragraph.

Apologies, this seems to be a problem with the wording. We meant that there are fewer synapses and that they are placed closer to the soma. We have rewritten this sentence to clarify (lines 254-257).

Did the immunohistochemistry analysis against PSD95 reveal an insight into the localization of excitatory synapses closer to PN somata as suggested in the previous review?

Since we are not suggesting that there is change in synapse placement, this point is moot.

Dye-coupling experiments:These seem inconclusive, and I am not convinced that the reviewer's concerns have been addressed.

We performed dye-coupling experiments with neurobiotin (323 Da) and serotonin

(177 Da). Neither probe could reveal the coupled partners of PNs when injected into PNs. When injected into non-PNs, we were able to label at most 2 coupled PNs. We attribute the low efficiency of dye coupling that we observed to the low unitary conductance of Gjd2b gap junction channels.

Zebrafish Gjd2b channels have a unitary conductance of around 24pS (Valiunas et al., 2004). This is also true of Cx36, the mammalian ortholog of Gjd2b, which have very small unitary conductances of 10-15pS (Srinivas et al., 1999). Cx36 has been shown to not support dye coupling (Teubner et al., 2000; Quesada et al., 2003). Further, when these same cells were made to overexpress Cx32, which has a larger conductance compared to Cx36, dye coupling could be observed (Quesada et al., 2003). Electrical coupling in the absence of dye coupling has been widely observed (Ransom and Kettenmann, 1990; Meda et al., 1991; Pérez-Armendariz et al., 1991; Moser, 1998; Teubner et al., 2000; Quesada et al., 2003). We would like to place these facts before the reviewers and want to suggest that the low conductance of Gjd2b channels might be the reason we are not able to observe dye coupling.

Presentation of DataThe authors carefully state that Gjd2b levels are reduced in morphants and mutants (e.g. line 131 or line 158), but this reduction (and not the complete absence) is difficult to understand if it is not mentioned initially that the anti-mouse Cx35/36 antibody cross reacts with Gjd2a and Gjd2b. This should be made clear from the beginning. Accordingly, line 116 should state "We confirmed that Gjd2a/b puncta..".

Already addressed above.

The authors state: "The scales for plots in Figures 1 and 2 have been matched.". I actually question that. First, I think that the Authors refer to Figures 1 and 3 as requested. The scale bar for Figure 3A, I doubt that is 10ms. I think it is 10 sec. Accordingly, in Figure 1A is 5pA, while in Figure 3A is 10pA. Finally, why is it so difficult to add the scale numbers in Figure3 as they have them in Figure 1 to compare the two figures directly? From this, I think the authors should be more careful regarding their statements and care more about their work presentation.

This was due to a misunderstanding on what the reviewer was asking us to do. We assumed that the reviewer was referring to all of the cumulative probability plots shown in Figures1 and 3 and we matched the scales on all the plots shown in Figures1 and 3. Representative traces are usually shown at a scale that best shows the detail in each case. Even so, as you will see from the scale bars, they are roughly of the same scale. The scale bar in 1A is roughly twice the size of 3A with twice the value. The 10ms label for 3A is a typo. We apologize and have now fixed it.

In the discussion (line 365-370) the authors emphasize their findings that glutamatergic synapses are decreased based on structural and physiological data but do not mention their contradictory finding with PSD95 immunohistochemistry. This should be discussed more cautiously.

We have removed the PSD95 data for reasons explained above.

"The authors present the holding potential as the membrane potential, which is confusing to distinguish between EPSPs and EPSCs." I referred to figure 4, where the authors in Panel B write the mV at the beginning of the traces that is a common practice for current-clamp recordings. In voltage-clamp recordings, there is the 0pA. The holding is usually added above, below, or in legend and refer as that (e.g., Holding at -65 mV). To this end, I also noticed that for Figures 1 and 3, any reference to holding potential is missing. Is that again the -65mV or different?

We have removed the holding potential label in the figure and mention it in the legends.

The authors should try to stimulate the inferior olive to specifically address the input of climbing fibers.The authors should discuss the discrepancy between their findings and the observation of gap junction coupling between cerebellar neurons recently found in adult zebrafish (Chang et al., 2020). That is important as in most neuronal networks; the gap junctions appear earlier, and they are more abundant in earlier developmental stages than the chemical synapses.

We do not view this as necessarily a discrepancy. Chang et al., in their 2020 PNAS paper (Chang et al., 2020) show that in adult zebrafish, PNs which share a chemical synaptic connection are also likely to be dye coupled and electrically coupled. The authors show Cx35/36 puncta in PNs but it is not known whether other connexins are also expressed in the same PNs. We did not observe dye coupling between PNs in the larva. As mentioned above, Cx35/36 have been shown to be not supportive of dye coupling and the dye coupling observed in the adult may be mediated via other connexins which are expressed by PNs in addition to Cx35. The composition of electrical synapses can change dramatically during development (Bhattacharya et al., 2019).

Reviewer #1:Reading the revised version of the manuscript "Gjd2b-mediated gap junctions promote glutamatergic synapse formation and dendritic elaboration in Purkinje neurons" by Sitaraman et al., I am pleased by the new experimental data and changes to the manuscript that have been added by the authors. The data is in most parts presented clearly, but a few points should be addressed to make some of their arguments more clear or to avoid misunderstandings of some of their claims.1) According to the guidelines of eLife the title of the manuscript should mention the use of zebrafish as model system e. g.: "Gjd2b-mediated gap junctions promote glutamatergic synapse formation and dendritic elaboration in zebrafish Purkinje neurons"

The policy states that the ‘biological system’ be mentioned where appropriate – which in our case is the Purkinje neuron.

2) The authors carefully state that Gjd2b levels are reduced in morphants and mutants (e.g. line 131 or line 158), but this reduction (and not the complete absence) is difficult to understand if it is not mentioned initially that the anti-mouse Cx35/36 antibody cross reacts with Gjd2a and Gjd2b. This should be made clear from the beginning. Accordingly, line 116 should state "We confirmed that Gjd2a/b puncta..".

Addressed above.

3) Line 221: the reduced number of synapses observed in TEM recordings could suggest but does not indicate a reduced number of formed synapses on PNs as this analysis is not able to distinguish between PNs and e.g. eurydendroid cells. I suggest to reword this sentence into "..these results indicate that Gjd2b is involved in the formation of synapses in the molecular layer of the cerebellum to which PN synapses significantly contribute.."

We have modified this sentence as suggested.

4) The authors report that anti-PSD95 immunohistochemistry analysis results in a higher number of synapse-counts on PNs that is contradictory to their results from TEM analysis and explain this with a limit in resolution by light microscopy (line 231) that does not allow one to unequivocally assign synapses to PNs. But should this erroneously assigning of synapses to PNs be the same in wild type and mutants? Why should this effect occur more often in the gjd2b mutants? This argument remains unclear to me.

Comment has been addressed above.

5) Line 240: the authors suggest that the increase in peak amplitude and faster kinetics of mEPSCs in gjd2b mutants results from excitatory synapses being placed closer to the PN soma, implying a stunted dendritic arbor and loss of distal synapses. This line of arguments is difficult for me to understand. Why should the loss of distal excitatory synapses lead to increases in peak amplitude?Or do the authors suggest that the stunted dendrites represent "compressed" dendrites in which synapses are moved closer to the soma? Then synapse density close to the soma should be investigated rather than distal synapses.Then the analysis reveals a reduced total dendritic branch length in gjd2b mutant PNs and confirms a stunted dendritic arbor from which the authors conclude that fewer synapses are placed proximally to the PN somata in gjd2b mutants (line 260). Is this not contradictory to the argument presented in line 240? For me this was confusing and I suggest to reword this paragraph. Did the immunohistochemistry analysis against PSD95 reveal an insight into the localization of excitatory synapses closer to PN somata as suggested in the previous review?

Also addressed above.

6) In the discussion (line 365-370) the authors emphasize their findings that glutamatergic synapses are decreased based on structural and physiological data but do not mention their contradictory finding with PSD95 immunohistochemistry. This should be discussed more cautiously.

We do believe that the combination of the electrophysiology and TEM data clearly point to a decrease in glutamatergic synapses. We present all of the caveats of the data in this paper in the supplementary table (Supplementary file 4).

7) Also, the authors should point out in the discussion that currently besides the cell type specific rescue of PN dendrite outgrowth in gjd2b mutants, they can currently not distinguish between cell-autonomous and non-cell autonomous effects of gjd2b loss on PNs. This should be made clear to the reader.

We have two ways in which we address the locus of action of Gjd2b:

1. Expressing full length Gjd2b in single PNs of mutants

2. Analysing the effect of Gjd2b puncta on individual branch behaviors in wild type PNs

These two experiments showed that signaling via Gjd2b locally can affect dendrite growth. These points are discussed in lines 448-456.

Reviewer #2:The manuscript is now improved in several aspects. The authors performed additional experiments to verify their observations. They followed most of the reviewers' recommendations, yet the overall impression is that the Authors did not consider some critical comments. Specifically, the ones that aimed to improve the data's presentation.Specifically:The authors state: "The scales for plots in Figures 1 and 2 have been matched.". I actually question that. First, I think that the Authors refer to Figures 1 and 3 as requested. The scale bar for Figure 3A, I doubt that is 10ms. I think it is 10 sec. Accordingly, in Figure 1A is 5pA, while in Figure 3A is 10pA. Finally, why is it so difficult to add the scale numbers in Figure3 as they have them in Figure 1 to compare the two figures directly? From this, I think the authors should be more careful regarding their statements and care more about their work presentation.

Addressed above.

Regarding the question of the specificity of the CF stimulation, the answer is not convincing. The fact that the authors used this approach in the past is not enough to claim specific. Why did the authors not try to stimulate the inferior olive?

We cannot stimulate the olive directly because it is located on the ventral side and is hard to reach with the stimulating electrode. The CF EPSCs are much larger than PF EPSCs, completely blocked by CNQX and reverse at around +10mV, all pointing to AMPAR type synaptic input. A thorough characterization of these inputs has been done in our 2015 paper (*eLife* 2015;4:e09158).

Regarding the Dye-coupling experiments where the authors state that they do not observe any dye coupling between PCs. In light of the new paper published in PNAS (Chang et al., 2020), the authors should discuss this discrepancy between their findings and what is observed in adult zebrafish. That is important as in most neuronal networks; the gap junctions appear earlier, and they are more abundant in earlier developmental stages than the chemical synapses.

Addressed above.

Regarding our previous comment: "The authors present the holding potential as the membrane potential, which is confusing to distinguish between EPSPs and EPSCs." I referred to figure 4, where the authors in Panel B write the mV at the beginning of the traces that is a common practice for current-clamp recordings. In voltage-clamp recordings, there is the 0pA. The holding is usually added above, below, or in legend and refer as that (e.g., Holding at -65 mV). To this end, I also noticed that for Figures 1 and 3, any reference to holding potential is missing. Is that again the -65mV or different?

Addressed above.

Recommendations for the authors:The manuscript is now improved in some aspects; however, the overall impression that I have is that the Authors did not take into consideration all the comments, and I found somehow quite lousy the revision of the previous manuscript. Specifically:The authors state: "The scales for plots in Figures 1 and 2 have been matched.". I actually question that. First, I think that the Authors refer to Figures 1 and 3 as requested. The scale bar for Figure 3A, I doubt that is 10ms. I think it is 10 sec. Accordingly, in Figure 1A is 5pA, while in Figure 3A is 10pA. Finally, why is it so difficult to add the scale numbers in Figure3 as they have them in Figure 1 to compare the two figures directly? From this, I think the authors should be more careful regarding their statements and care more about their work presentation.Regarding the question of the specificity of the CF stimulation, the answer is not convincing. The fact that the authors used this approach in the past is not enough to claim specific. Why did the authors not try to stimulate the inferior olive?Regarding the Dye-coupling experiments where the authors state that they do not observe any dye coupling between PCs. In light of the new paper published in PNAS (Chang et al., 2020), the authors should discuss this discrepancy between their findings and what is observed in adult zebrafish. That is important as in most neuronal networks; the gap junctions appear earlier, and they are more abundant in earlier developmental stages than the chemical synapses.Regarding our previous comment: "The authors present the holding potential as the membrane potential, which is confusing to distinguish between EPSPs and EPSCs." I referred to figure 4, where the authors in Panel B write the mV at the beginning of the traces that is a common practice for current-clamp recordings. In voltage-clamp recordings, there is the 0pA. The holding is usually added above, below, or in legend and refer as that (e.g., Holding at -65 mV). To this end, I also noticed that for Figures 1 and 3, any reference to holding potential is missing. Is that again the -65mV or different?Reviewer #3:As outlined in my original review of this manuscript, I view it as interesting and potentially impactful, but preliminary in its interpretations and not consistently convincing. I do not view this has having changed in a meaningful way with these revisions. The new data provided do not particularly strengthen conclusions, and in one case conflict with the manuscript's narrative. Most reviewers' comments have been addressed without further experiments, but not in a way that is consistently satisfying. I address these revisions below, broken down by sections of the rebuttal letter.Morphants:I remain unconvinced that a conclusive negative result is shown for PPR in Figure 4. There is a nonsignificant drop in PPR in the morphants and a sometimes-significant rise in PPR at short ISIs in the mutants. I understand that the distributions overlap, and that this is why most results are not significant, but I do not believe that the results and the experimental n are sufficient to support the claim that this is a negative result. I do not believe that this argument has changed appreciably since the initial submission.

Addressed above.

I accept the authors' assertion that increased peak amplitude for the mutant is convincingly demonstrated in Figure 3D.I continue to think that the mutant provides stronger support for the authors' claims than the MOs do.

We also agree, which is why we made the mutants after initial promising results with the morphants.

TALEN Mutants:I am generally convinced by the authors' responses to these questions and with the added data, although I defer to the reviewer who originally raised the issue of the two orthologs with regard to whether this has been adequately addressed.Electrophysiological recordings:As described above, I continue not to be convinced of a negative result for the PPD data shown in Figure 4D-G.

Addressed above.

The arguments presented about blockers and the new NMDA pulse data appear to be valid to me, but I defer to the relevant reviewer.EM Analysis:I understand the converging lines of evidence that the authors refer to (ephys and EM), and understand that these could be viewed as complementary, given each line's strengths and caveats. This does not really address the reviewer's concern, although I defer to him/her with regard to whether they are convinced.

Thank you. We are not aware of the concerns regarding the converging lines of evidence. The concerns regarding EM data are addressed above.

The new PSD95 results are concerning. They are among the strongest results in the paper, and while the authors give two caveats associated with interpreting these data, the overall picture is not, on the whole, supportive of their interpretations.I do not see that fish-level data have been added to Figure 5, where box plots continue to show data from hundreds of micrographs drawn from a small number of fish.Dye-coupling experiments:These seem inconclusive, and I am not convinced that the reviewer's concerns have been addressed.

All addressed above.

CaMKII-signalling:This was an interesting but not fully supported element of the original manuscript. I agree with the decision to withdraw these data, but it leaves a less impactful paper.

We are flummoxed as we removed these data at the suggestion of the reviewers. We hope that suggestion was made after evaluating the rest of the paper on its own strengths. We have now included the CaMKII data in the manuscript.

References:

Bhattacharya A, Aghayeva U, Berghoff EG, Hobert O. 2019. Plasticity of the Electrical Connectome of *C. elegans*. *Cell***176**:1174-1189.e16.

doi:10.1016/j.cell.2018.12.024

Chang W, Pedroni A, Hohendorf V, Giacomello S, Hibi M, Köster RW, Ampatzis K. 2020. Functionally distinct Purkinje cell types show temporal precision in encoding locomotion. *PNAS***117**:17330–17337.

doi:10.1073/pnas.2005633117

Meda P, Chanson M, Pepper M, Giordano E, Bosco D, Traub O, Willecke K, el Aoumari A, Gros D, Beyer EC. 1991. in vivo modulation of connexin 43 gene expression and junctional coupling of pancreatic B-cells. *Exp Cell Res*

**192**:469–480. doi:10.1016/0014-4827(91)90066-4

Moser T. 1998. Low-conductance intercellular coupling between mouse chromaffin cells in situ. *The Journal of Physiology***506**:195–205.

doi:https://doi.org/10.1111/j.1469-7793.1998.195bx.x

Pérez-Armendariz M, Roy C, Spray DC, Bennett MV. 1991. Biophysical properties of gap junctions between freshly dispersed pairs of mouse pancreatic β cells.

*Biophys J***59**:76–92. doi:10.1016/S0006-3495(91)82200-7

Quesada I, Fuentes E, Andreu E, Meda P, Nadal A, Soria B. 2003. On-line analysis of gap junctions reveals more efficient electrical than dye coupling between islet cells. *American Journal of Physiology-Endocrinology and Metabolism*

**284**:E980–E987. doi:10.1152/ajpendo.00473.2002

Ransom BR, Kettenmann H. 1990. Electrical coupling, without dye coupling, between mammalian astrocytes and oligodendrocytes in cell culture. *Glia*

**3**:258–266. doi:10.1002/glia.440030405

Srinivas M, Rozental R, Kojima T, Dermietzel R, Mehler M, Condorelli DF, Kessler JA, Spray DC. 1999. Functional properties of channels formed by the neuronal gap junction protein connexin36. *J Neurosci***19**:9848–9855.

Teubner B, Degen J, Söhl G, Güldenagel M, Bukauskas FF, Trexler EB, Verselis VK, De Zeeuw CI, Lee CG, Kozak CA, Petrasch-Parwez E, Dermietzel R, Willecke K. 2000. Functional expression of the murine connexin 36 gene coding for a neuron-specific gap junctional protein. *J Membr Biol***176**:249–262.

doi:10.1007/s00232001094

Valiunas V, Mui R, McLachlan E, Valdimarsson G, Brink PR, White TW. 2004. Biophysical characterization of zebrafish connexin35 hemichannels. *Am J*

*Physiol Cell Physiol***287**:C1596-1604. doi:10.1152/ajpcell.00225.2004

[Editors' note: further revisions were suggested prior to acceptance, as described below.]

Essential Revisions:Please modify the text of the paper to identify caveat, alternate interpretations and open questions for future research, as suggested in the reviewers comments, below.

Thank you for the suggestions below. We have modified the text and included new analysis as suggested by the reviewers.

Reviewer #1 (Recommendations for the authors):1. It would be more clear to name the splice-blocking morpholino as 'Gjd2b-MO'

All references to the splice blocking morpholino are now mentioned as Gjd2b-MO in the manuscript.

2. Control morpholinos were tested separately in experiments using immunolabeling, but were not used in the electrophysiology experiments. This is unusual but acceptable.

Since we showed that the control morpholino did not alter protein levels as observed with immunolabeling (figure 1—figure supplement 2E), we did not perform further electrophysiology experiments.

3. Figure 1. Supplement 1. Please label columns and rows in panel B.

We have added the labels to panel B of Figure 1—figure supplement 1.

4. For morphants, please state if experimental and control animals were from the same clutch and how many clutches of embryos were used for each experiment.

For immunohistochemistry, morphant and control animals were from the same clutch and the processing of control and morphant embryos was done in parallel. Data were collected from two such clutches of embryos.

For the mEPSC data, since parallel processing was not possible, control and morphant embryos were from different clutches. Control data are from 7 cells recorded from 5 clutches and morphant data are from 12 cells from 8 clutches. This information is now added to the respective figure legends.

5. Figure 4. Panel B, list specific ages in legend, rather than 'mixed ages'. Panel C, state that this is done on the presence of TTX.

The following was added to the figure legend:

EPSCs recorded at a hyperpolarized holding potential (bottom row traces) and at a depolarized holding potential (top traces) in normal saline (left side traces) and in saline containing the AMPAR blocker CNQX (right side traces). No EPSCs were detected in the presence of CNQX at -65 or +60mV at 7dpf (N = 4 cells) or at 19dpf (N = 2 cells). Data from 19 dpf shown above.

We mentioned in the methods that the NMDAR currents were recorded in the presence of TTX. We have now added this sentence to figure 4 legend as well.

6. In the first paragraph of discussion, authors should more clearly state that they think that conductance through Cx36/Gjd2b-mediated gap junctions is not be sufficient to pass dye and they are therefore not able to identify cells coupled to PCs.

We have addressed it thus (line 361):

“We found Gjd2b puncta localized to PN cell membrane but when dye was injected into PNs, we failed to observe any dye-coupled cells. This may be attributable to the low conductance of zebrafish Gjd2b channels, which have a unitary conductance of around 24pS (Valiunas et al., 2004).

7. Given the ambiguity about the sequence of events governing branch extension gap junction formation -> synaptogenesis->branch extension, repeat versus gap junction formation -> branch extension-> synaptogenesis, repeat., the interpretation on line 242 'resulting in fewer synapses' is overstated.

We have replaced “resulting in” with “having”. (line 254)

8. The authors state that it is likely that gap junctions promote dendritic arbor growth directly, independent of their actions on chemical synapses. What evidence (citation) supports this statement?

This statement is based on our data in Figure 7C where we saw increased elongations of branches containing at least one Gjd2b punctum compared to those that didn’t have any. These differences were observed even within the 5-minute imaging windows. We have moved this sentence ahead (line 442) to explain the context better.

9. On pg 19 the first full paragraph is repeated in the next section.

This was an oversight on our part. It has been rectified.

10. Figure 7, supplement 1. There is a problem with the labeling

This has been fixed.

Reviewer #2 (Recommendations for the authors):Electrophysiology data:1. The data on mEPSC frequency is clear evidence for fewer excitatory synapses. However, interpreting larger amplitudes and faster decays as an explanation for the loss of distal synapses is problematic. The shift towards larger amplitudes can also be explained simply by the absence of weaker synapses rather than arguing for the absence of distal synapses. Moreover, larger amplitudes can also be a homeostatic scaling mechanism to compensate for fewer synapses. Regarding the decay, this evidence would be more compelling if these experiments were done in current clamp where the absence of gap junctions would lead to longer voltage decay as a result of increased input resistance of neurons. In contrast, the faster current decay observed by the authors could also be a result of different subunit composition or other biophysical considerations (Laurence et al., Nature Neuroscience, 2005; Kumar et al., JNeuroscience 2002), and unless that is ruled out explicitly, I would recommend adding this as an important alternative explanation in the discussion.

We thank the reviewer for their appreciation of the mEPSC frequency data. Regarding the interpretation of the amplitude and kinetics data, we did not interpret them as indicative of loss of distal synapses. We only offer these observations as a motivation for looking into the dendritic arbor structure of mutant PNs (line 226-234). Nevertheless, we have added the following sentences to the discussion (line: 397):

“The amplitude and kinetics of mEPSCs of mutant PNs followed a trend that was consistent with smaller dendritic arbors. However, alternate explanations such as changes in receptor numbers and subunit types are also possible. “

2. What is the input resistance of mutant neurons versus wildtype Purkinje neurons? Given the smaller dendritic arbors and the absence of gap junctions, one would expect Gjdb2-/- neurons have a higher input resistance. It will be good to see these data.

We thank the reviewer for bringing up this interesting point. We compared the input resistance of wild type and mutant Purkinje neurons. Interestingly, the input resistances were not different when recorded with potassium gluconate internal solution but when recorded with cesium gluconate internal solution, the input resistance was significantly higher in the mutants. This observation is also consistent with the smaller arbor size. With potassium gluconate, probably only the proximal dendrites contribute to the leak, while with cesium gluconate, a larger extent of dendritic arbors contribute to the leak and therefore to the input resistance. A smaller dendritic arbor in the mutants would mean smaller dendritic leak, and hence larger input resistance. We have now included these data in Figure 3—figure supplement1.

3. In comparing the WT data in Figure 1D,E,F with Figure 3D,E,F, I am confused by differences in the distribution of WT data between these figures. There is a long tail in the distribution of WT data in Figure 1D,E,F which is absent in Figure 3D,E,F. I am curious why this is the case.

The long tail in Figure 1 D, E, and F is due to a few events, which are on the right end of the distribution. This can be seen in the inset box plots as well. We looked at whether these large values were contributed by one or two cells and this was not the case. The large values were typically from few events (2-3) in almost every cell recorded from. The morphant and mutant data sets were also acquired with a time gap of several months in between. The corresponding wild type data sets were also recorded several months apart. So, the few large outlier values seen only in Figure 1 may also be related to fish population level variability.

EM data:1. I agree with the other reviewers that while the EM data are beautiful and hard to collect and analyze, not being able to attribute synapses to PNs significantly limits the conclusions one can reach from this experiment. I do not know much about cerebellar circuitry but is there any estimate for what fraction of excitatory synapses are formed on PNs versus other neurons? Also, the inability to distinguish between excitatory and inhibitory synapses is a major limitation. Between the EM and electrophysiology data, I would argue that the electrophysiology data are far stronger. I understand that the authors see this as converging evidence, but in my opinion, the EM data have substantial caveats and at best provide weak support for the conclusions the authors are trying to reach. If, for instance, the authors have data on spontaneous IPSCs in mutant and WT neurons that are similar in frequency, that can at least help argue that the numbers of inhibitory synapses are similar.

We agree with the reviewer regarding these caveats and have stated them as such in the Supplementary file 4 document. Unfortunately estimates of synapse numbers within the molecular layer of zebrafish cerebellum are not available. We have not recorded mIPSCs from these neurons as this would entail substantial new effort. We plan to investigate the effect of gjd2b on inhibition in the future.

2. I have not seen the PSD95 data, but I agree with the authors that PSD95 staining is not a compelling experiment. Light microscopy resolution is a major challenge. If anything, the authors could have tried expressing PSD95-tagged GFP in individual neurons in mutant and WT fish as was done in Niell et al. (Nature Neuroscience 2004) in the zebrafish optic tectum. I am not suggesting that the authors do this experiment but wanted to just throw in support for their argument that PSD95 staining is inconclusive.

We performed PSD95 immunohistochemistry to tag the endogenous PSD95. We did not try overexpressing PSD95-EGFP as multiple studies indicate that the overexpression of PSD95 in neurons affects synaptogenesis and dendrite growth (Graf et al., 2004; Charych et al., 2006; Nikonenko et al., 2008) to name a few. Thank you for agreeing with us regarding this experiment.

Dendritic elaboration and CaMKII1. The gain of function experiments are interesting, but perhaps I am missing something here. My understanding is that functional electrical synapses need the assembly of a pore in the presynaptic and post-synaptic neuron. So, how would expressing Gjd2b in one neuron ensure functional electrical connectivity with other neurons? I see this is addressed in the limitations document, but the authors' reliance on pore dead experiments is unconvincing. If anything, the pore dead neurons have dendrites that are comparable to WT neurons and longer than the Gjd2b neurons (Figure 7b). I think the authors need to do a statistical analysis of differences between WT and Gjd2b rescue as well as between WT and Gjd2b pore-dead mutants. I think there is something interesting there that might allude to functional aspects of non-pore forming regions of Gjd2b. In the absence of clear experiments to demonstrate functional electrical synapses, I think this experiment falls significantly short of implicating electrical synapses in dendrite elaboration. If the authors do want to make this claim, in the very least they need to show that the "rescued" neurons have comparable excitatory synapses

We had technical difficulty in performing mEPSC recordings on rescued neurons as it was difficult to ascertain Gjd2b-mCherry expression at the wide-field microscope in our electrophysiology rig and the expressing neurons also tended to be deeper than could be accessed by our recording pipette. The dendrites of pore-dead expressing neurons are not significantly longer than Gjd2b neurons (p=0.44). The point about functional electrical synapses in the rescue experiment is addressed in point #3 above. We also show that in wild type neurons, when Gjd2b is expressed, branches containing one or more Gjd2b puncta elongate more than those that don’t have any within 5 minute imaging windows. This is another piece of evidence in support of Gjd2b promoting dendritic growth. However, as stated above, we cannot completely disambiguate dendrite growth promoting versus synaptogenic roles as the primary effect of Gjd2b. We have included this point in the discussion (line 440-448).

2. A lot of the work on dendritic elaboration has parallels with the rich body of work done in Hollis Cline's lab which the authors reference extensively. However, it's not clear to me that the dendritic effects are not simply a downstream consequence of the absence of Gjd2b rather any information transmitted through the electrical synapses. Since Gjd2b knockout reduces the number of AMPAR synapses based on the electrophysiology, isn't a simple explanation for all the dendritic effects simply a consequence of fewer AMPAR synapses as shown in Haas et al. (PNAS, 2006). Moreover, given the lack of direct evidence that the rescue experiments lead to functional electrical synapses, I am not convinced that molecules transmitted through gap junctions are somehow responsible for elevated CaMKII.

These points have also been brought up by Reviewer #1. We include in the discussion the two possible scenarios (Gjd2b→ AMPAR synapses → dendritic growth and Gjd2b→ dendritic growth → AMPAR synapses) mentioned by Reviewers 1 and 2. In addition we have removed strong wordings of causality with respect to CaMKII as suggested by both the reviewers.

In summary, while this paper represents a substantial amount of work and relies on converging lines of evidence to arrive at their conclusions, there are several limitations within each technique and these shortcomings are not addressed by the complimentary experiments.The authors present good evidence for fewer chemical synapses and shorter dendrites in Purkinje neurons in fish where Gjd2b dependent electrical synapses are knocked down or knocked out. The concerns about electrophysiology data can be addressed in the discussion as an important caveat.

We have now stated these caveats in the discussion (line 397-401).

However, my bigger concern is with disambiguating Gjd2b mediated changes in dendritic structure from downstream effects of simply having fewer AMPAR synapses. Previous work has provided compelling evidence that chemical transmission through AMPAR synapses is a key driver of dendritic elaboration. So, if there are fewer AMPAR synapses, is it not unsurprising that the dendrites are smaller and that has nothing to do directly with Gjd2b function? Perhaps I am missing a key piece of the argument here and would be happy to be proven wrong.

This point is addressed above.

Reviewer #3 (Recommendations for the authors):Having read through the previous response to the reviewers, I think that the authors have addressed them very well. I do not think further experiments or analyses are required.

Reference:

Charych EI, Akum BF, Goldberg JS, Jörnsten RJ, Rongo C, Zheng JQ, Firestein BL. 2006. Activity-independent regulation of dendrite patterning by postsynaptic density protein PSD-95. J Neurosci 26:10164–10176. doi:10.1523/JNEUROSCI.2379-06.2006

Graf ER, Zhang X, Jin S-X, Linhoff MW, Craig AM. 2004. Neurexins Induce Differentiation of GABA and Glutamate Postsynaptic Specializations via Neuroligins. Cell 119:1013–1026. doi:10.1016/j.cell.2004.11.035

Haas K, Li J, Cline HT. 2006. AMPA receptors regulate experience-dependent dendritic arbor growth in vivo. Proc Natl Acad Sci USA 103:12127–12131. doi:10.1073/pnas.0602670103

Miller AC, Whitebirch AC, Shah AN, Marsden KC, Granato M, O’Brien J, Moens CB. 2017. A genetic basis for molecular asymmetry at vertebrate electrical synapses. *eLife* Sciences 6:e25364. doi:10.7554/*eLife*.25364

Nikonenko I, Boda B, Steen S, Knott G, Welker E, Muller D. 2008. PSD-95 promotes synaptogenesis and multiinnervated spine formation through nitric oxide signaling. J Cell Biol 183:1115–1127. doi:10.1083/jcb.200805132